# A synthetic growth switch based on controlled expression of RNA polymerase

Jérôme Izard[1,2,†], Cindy DC Gomez Balderas[1,2,†], Delphine Ropers[2], Stephan Lacour[1,2], Xiaohu Song[3], Yifan Yang[3], Ariel B Lindner[3], Johannes Geiselmann[1,2,*,‡] & Hidde de Jong[2,**,‡]

## Abstract

The ability to control growth is essential for fundamental studies of bacterial physiology and biotechnological applications. We have engineered an *Escherichia coli* strain in which the transcription of a key component of the gene expression machinery, RNA polymerase, is under the control of an inducible promoter. By changing the inducer concentration in the medium, we can adjust the RNA polymerase concentration and thereby switch bacterial growth between zero and the maximal growth rate supported by the medium. We show that our synthetic growth switch functions in a medium-independent and reversible way, and we provide evidence that the switching phenotype arises from the ultrasensitive response of the growth rate to the concentration of RNA polymerase. We present an application of the growth switch in which both the wild-type *E. coli* strain and our modified strain are endowed with the capacity to produce glycerol when growing on glucose. Cells in which growth has been switched off continue to be metabolically active and harness the energy gain to produce glycerol at a twofold higher yield than in cells with natural control of RNA polymerase expression. Remarkably, without any further optimization, the improved yield is close to the theoretical maximum computed from a flux balance model of *E. coli* metabolism. The proposed synthetic growth switch is a promising tool for gaining a better understanding of bacterial physiology and for applications in synthetic biology and biotechnology.

**Keywords** bacterial physiology; growth; RNA polymerase; synthetic biology; biotechnology
**Subject Categories** Synthetic Biology & Biotechnology; Quantitative Biology & Dynamical Systems; Metabolism
**Mol Syst Biol.** (2015) 11: 840

## Introduction

Optimizing growth in a given environment is a major challenge for microorganisms. In particular, changes in nutrient availability require a rapid adjustment of growth rate and therefore the reallocation of available resources to cellular functions. Early work in bacterial physiology has characterized the macromolecular composition of microbial cells as a function of growth rate (Schaechter *et al*, 1958; Kjeldgaard 1961; Neidhardt & Fraenkel, 1961; Bremer & Dennis, 1996; reviewed in Maaløe & Kjeldgaard, 1966; Scott & Hwa, 2011). These experiments have shown that the DNA, RNA, and protein contents are correlated with the growth rate and not the nature of the specific nutrients available for growth. The culture history also plays a role, especially when the nutritional quality of the medium is low (Ehrenberg *et al*, 2013). Other physiological quantities that vary with the growth rate include the intracellular concentrations of RNA polymerase and ribosome, the mass ratio of total protein and RNA, and the size of metabolic pools (Bremer & Dennis, 1996). Recently, there has been a regained interest in the role of global cell physiology, notably the activity of the transcriptional and translational machinery, in the control of gene expression, both in balanced growth and during growth transitions (Klumpp *et al*, 2009; Levy & Barkai, 2009; Molenaar *et al*, 2009; Scott *et al*, 2010; Goelzer & Fromion, 2011; Karr *et al*, 2012; Berthoumieux *et al*, 2013; Ehrenberg *et al*, 2013; Gerosa *et al*, 2013; Keren *et al*, 2013; O'Brien *et al*, 2013; You *et al*, 2013; de Lorenzo, 2014; Scott *et al*, 2014; Weisse *et al*, 2015).

The ability to experimentally control the growth rate is thus crucial for studying bacterial physiology. It is also of central importance for applications in biotechnology, where often the goal is to limit or even arrest growth. Growth-arrested cells with a functional metabolism open the possibility to channel resources into the production of a desired metabolite, instead of wasting nutrients on biomass production (Flickinger & Rouse, 1993; Sonderegger *et al*, 2005). A variety of approaches have been classically employed to limit growth. One example is the use of antibiotics targeting the transcription or translation machinery or DNA topology. This

1 Université Grenoble Alpes, Laboratoire Interdisciplinaire de Physique (CNRS UMR 5588), Saint Martin d'Hères, France
2 INRIA, Grenoble – Rhône-Alpes research center, Saint Ismier, France
3 Center for Research and Interdisciplinarity, INSERM U1001, Medicine Faculty, Site Cochin Port-Royal, University Paris Descartes, Paris, France
*Corresponding author. Tel: +33 476514753; E-mail: hans.geiselmann@ujf-grenoble.fr
**Corresponding author. Tel: +33 476615335; E-mail: hidde.de-jong@inria.fr
†These authors contributed equally to this work
‡These authors contributed equally to this work

approach can lead to cell death, however, and in addition to being too costly in an industrial setting, mutants resistant to antibiotics quickly emerge and take over the population. An alternative strategy consists in imposing limitations on nutrients essential for growth, in particular carbon, nitrogen, and phosphate sources. This intervention strongly affects metabolism though, as cells adjust their flux distribution and enzyme levels to nutrient limitations. As a consequence, the engineered metabolic pathways and genetic circuits may not function or function suboptimally.

The above limitations call for novel strategies to limit or completely stop growth. Ideally, it should be possible to grow a cell population to a certain biomass and then switch off growth. The enzymes present at the time of growth arrest are expected to remain functional, thus enabling high-yield production of a metabolite of interest. When the degradation of enzymes and other proteins threatens the stability of metabolism, it should be possible to switch on growth again, thus alternating phases of biomass accumulation and product synthesis. In addition to being reversible, the exercised control over the growth rate should be medium independent, in the sense of being applicable over a wide range of medium compositions and corresponding nutrient uptake patterns.

Here, we present a synthetic growth switch that satisfies the above criteria. We report the construction of an *Escherichia coli* strain in which the transcription of the operon encoding two large subunits of RNA polymerase, $\beta$ and $\beta'$, is under the tight control of an isopropyl-$\beta$-D-1-thiogalactopyranoside (IPTG)-inducible promoter. Previous attempts at constructing such a strain have failed to give conclusive results (Nomura *et al*, 1987). The challenge has been abandoned after the heyday of research in bacterial transcription, but modern techniques of chromosome engineering have allowed us to successfully tackle this endeavor. In our engineered strain, the growth rate can be switched on and off by changing the inducer concentration in the medium and thus the concentration of RNA polymerase. The system is purely genetic and therefore does not depend on the composition of the medium, apart from the supply of inducer. When the production of RNA polymerase is completely shut off, the growth rate decreases continuously, but cells remain metabolically active and do not die. The analysis of single cells in a microfluidics device shows that growth control is reversible: a growth-arrested culture resumes normal growth after the inducer, IPTG, is added back to the medium. Our data suggest that the switch between growth and growth arrest is due to the highly ultrasensitive response of the growth rate to a change in concentration of the $\beta$ and $\beta'$ subunits, as quantified by means of a fluorescent tag on $\beta'$. This indicates that the variation of *rpoBC* expression is the effective cause of the change in growth rate.

In order to provide a proof-of-principle of the applicability of our growth switch in a biotechnological context, we endow our *E. coli* strain with the capacity to convert glucose to glycerol, by expressing appropriate yeast enzymes. We measure extracellular levels of glucose and glycerol and show that this allows the production of glycerol during growth arrest, at a yield much higher than that obtained in a growing wild-type strain expressing the same enzymes. Remarkably, the glycerol production yield is close to the maximum theoretical yield computed from a flux balance model of *E. coli* metabolism. These results demonstrate that the desired reallocation of resources from growth toward the production of a molecule of interest is feasible and indeed highly profitable.

In summary, we present a growth switch that is a generic tool of broad interest for applications in synthetic biology and biotechnology. Since the controller is a genetic element, our system is modular: the growth switch can be interfaced with any other (natural or synthetic) cellular network via intracellular signals. Moreover, the principle underlying its design is applicable to other bacteria with a single RNA polymerase.

# Results

### Reengineering the control of RNA polymerase expression

The core RNA polymerase of *Escherichia coli* consists of three different subunits and has the composition $\alpha_2\beta\beta'$. The $\alpha$ subunit being produced in excess during growth (Hayward & Fyfe, 1978), the amount of the $\beta\beta'$ subunits sets the overall concentration of RNA polymerase (Engbaek *et al*, 1976; Kawakami *et al*, 1979; Shepherd *et al*, 2001). These two subunits are the product of the *rpoB* and *rpoC* genes, organized in an operon with upstream genes coding for ribosomal proteins (*rplKAJL*) (Yamamoto & Nomura, 1978). The expression of *rpoBC* genes is regulated both at the transcriptional and translational level by complex and still incompletely understood mechanisms. An attenuator of transcription located in the intergenic region *rplL-rpoB* decouples the expression of ribosomal proteins and the two RNA polymerase subunits (Ishihama & Fukuda, 1980; Dykxhoorn *et al*, 1996).

Growth of a bacterial cell requires the duplication of its contents, in particular protein, RNA, and DNA which make up 75–90% of cellular material (Bremer & Dennis, 1996). Synthesis of RNA and protein starts with transcription, so the production of RNA polymerase is essential for growth and survival of the cell. When the activity of RNA polymerase is inhibited, for example, by the action of an antibiotic such as rifampicin, the bacteria stop growing (Campbell *et al*, 2001). Since *E. coli* possesses only one RNA polymerase, this suggests that, by adjusting the concentration of the limiting $\beta$ and $\beta'$ subunits, we can vary the growth rate of the cell.

We therefore replaced the natural promoter of *rpoBC* of the *E. coli* strain BW25113 by an IPTG-inducible promoter. The transcription of *rpoBC* was completely isolated from the upstream ribosomal proteins by introducing a selection cassette and a strong transcriptional terminator (Fig 1A and Appendix Fig S1). Two extra copies of the Lac repressor gene were added to the chromosome in two different loci in order to prevent the relief of repression by mutations inactivating the single original copy of *lacI* (Fig 1B and Appendix Fig S2). Whereas a bacterial population (typically between $10^6$ and $10^9$ bacteria in a growth experiment) most likely contains an individual carrying a mutation in one copy of *lacI*, a double mutation that inactivates both copies of the gene is very unlikely. Hereafter, we call the resulting strain with reengineered transcriptional control of RNA polymerase expression the "R strain" and the wild-type strain from which it has been derived the "W strain".

### Reengineering transcriptional control of RNA polymerase allows a medium-independent growth switch

Does the strain with inducible transcriptional control of the RNA polymerase $\beta$ and $\beta'$ subunits indeed allow the growth rate to be

**A**

**B**

**Figure 1.   Construction of an *E. coli* strain with inducible expression of the *rpoBC* genes encoding the $\beta\beta'$ subunits of RNA polymerase.**

A   We have replaced the *rpoBC* promoter region by an IPTG-inducible promoter. A strong transcriptional terminator, *rrnBt1*, was inserted upstream of the selection cassette (*spcR*) and the T5 promoter with two *lac* operator sequences. *rpoBC* transcription is thus efficiently repressed by the *lac* repressor LacI (Lutz & Bujard, 1997).

B   The gene encoding LacI is present in three copies on the chromosome, one natural copy (in black) and two additional copies (in red). The engineered strain is referred to as "R", in distinction to the wild-type strain "W". For more details on the construction, see Appendix Figs S1 and S2.

controlled? In order to answer this question, we characterized the R strain in different growth media, adjusting *rpoBC* expression by varying the IPTG concentration.

The R and W strains were incubated on LB agar plates with a high concentration of IPTG (1,000 μM) and a single colony was picked from the plates and precultured in M9 minimal medium, supplemented with 0.2% glucose and 1,000 μM IPTG to ensure maximal production of RNA polymerase. The overnight precultures were centrifuged, washed, and redituted into 96-well microplates containing fresh M9 minimal medium with 0.2% glucose and different concentrations of IPTG (Materials and Methods). The microplate culture conditions were chosen so as to improve aeration (use of glass beads for stirring, high frequency of shaking) and obtain the same growth rate as in shake flasks (see below).

Figure 2A shows the growth curves of the R strain obtained for a range of IPTG concentrations, varying from 0 to 1,000 μM. As a control, we also show the growth curve of the W strain. The growth curves are identical for all concentrations of IPTG during the first 3 h of the experiment because there is enough RNA polymerase from the overnight preculture to ensure a maximal growth rate. After this initial growth phase determined by the conditions of the preculture, the concentration of IPTG controls growth of the R strain in a dose-dependent manner. To a first approximation, the growth curves can be divided into two categories. In the first category, for IPTG concentrations of 30 μM and higher, growth is normal or close to normal, as compared to the wild-type strain, and the cultures reach stationary phase at an absorbance of 0.5, once all nutrients in the medium have been exhausted. The bacteria first consume all glucose in the medium and then continue growth at a lower rate, utilizing by-products secreted during growth on glucose, notably acetate (Andersen & von Meyenburg, 1980; El-Mansi & Holms, 1989; Wolfe, 2005). For the highest concentrations of IPTG (100 and 1,000 μM), the growth kinetics of the R strain is practically indistinguishable from that of the W strain. In the second category, for IPTG concentrations of 20 μM and lower, growth stops after a few hours at an absorbance level well inferior to that reached in the wild-type strain.

In order to quantify these observations, we computed the growth rate attained for the different concentrations of IPTG at an appropriate stage of the growth kinetics. More precisely, for concentrations of 20 μM and lower, we computed the growth rate after

growth arrest at low absorbance (beyond 1,000 min). For concentrations of 30 μM and higher, we considered the time window where the absorbance is larger than 0.05 (after 5–6 generations, when RNA polymerase from the preculture has been strongly diluted out) and smaller than 0.2 (above which growth is no longer exponential, due to growth-limiting oxygen transfer rates). In every case, we fitted an exponential function to the data, typically consisting of several dozens of absorbance measurements, to obtain a precise estimate of the growth rate. The results in Fig 2B provide a quantitative picture of the observed growth switch. At IPTG concentrations of 20 μM and lower, the growth rate is 0, while between 20 and 30 μM it jumps to values beyond 0.008 min$^{-1}$. For the highest IPTG concentrations, the growth rate takes a value of $0.012 \pm 0.0008$ min$^{-1}$, corresponding to a doubling time of 57 min, characteristic for the wild-type BW25113 *E. coli* strain (Volkmer & Heinemann, 2011).

For IPTG concentrations of 30 μM and higher, we repeated the experiment with a 10-fold higher dilution ratio of the preculture, so as to make sure that the observed depletion of the nutrients in the medium, and the corresponding growth rate, is really due to the continued synthesis of RNA polymerase at the specified IPTG concentration and not a residual effect from the preculture (Fig EV1). We also compared the growth rate computed from the data obtained from microplate experiments with that obtained in experiments in shake flasks and observed excellent agreement (Fig 2B). This assures that the growth conditions in the microplate, at least for the bacterial population densities considered here, are sufficiently similar to growth conditions in shake flasks. Furthermore, we tested that the growth rate of the W strain did not vary with the IPTG concentrations, showing that the effect of IPTG on the growth rate is really due to transcription regulation of the *rpoBC* genes and not to some other side-effect (Fig EV1).

The results obtained with the R strain grown in minimal M9 medium with glucose suggest that for low concentrations of IPTG, the quantity of newly synthesized $\beta$ and $\beta'$ subunits is not enough to sustain growth, whereas growth reaches the maximum rate for higher concentrations of IPTG. The question can be asked to which extent the observed switching behavior depends on the specific induction system used. It is well known that the response of IPTG-dependent promoters to the inducer concentration in the medium is cooperative (Kuhlman *et al*, 2007; Robert *et al*, 2010), in large part

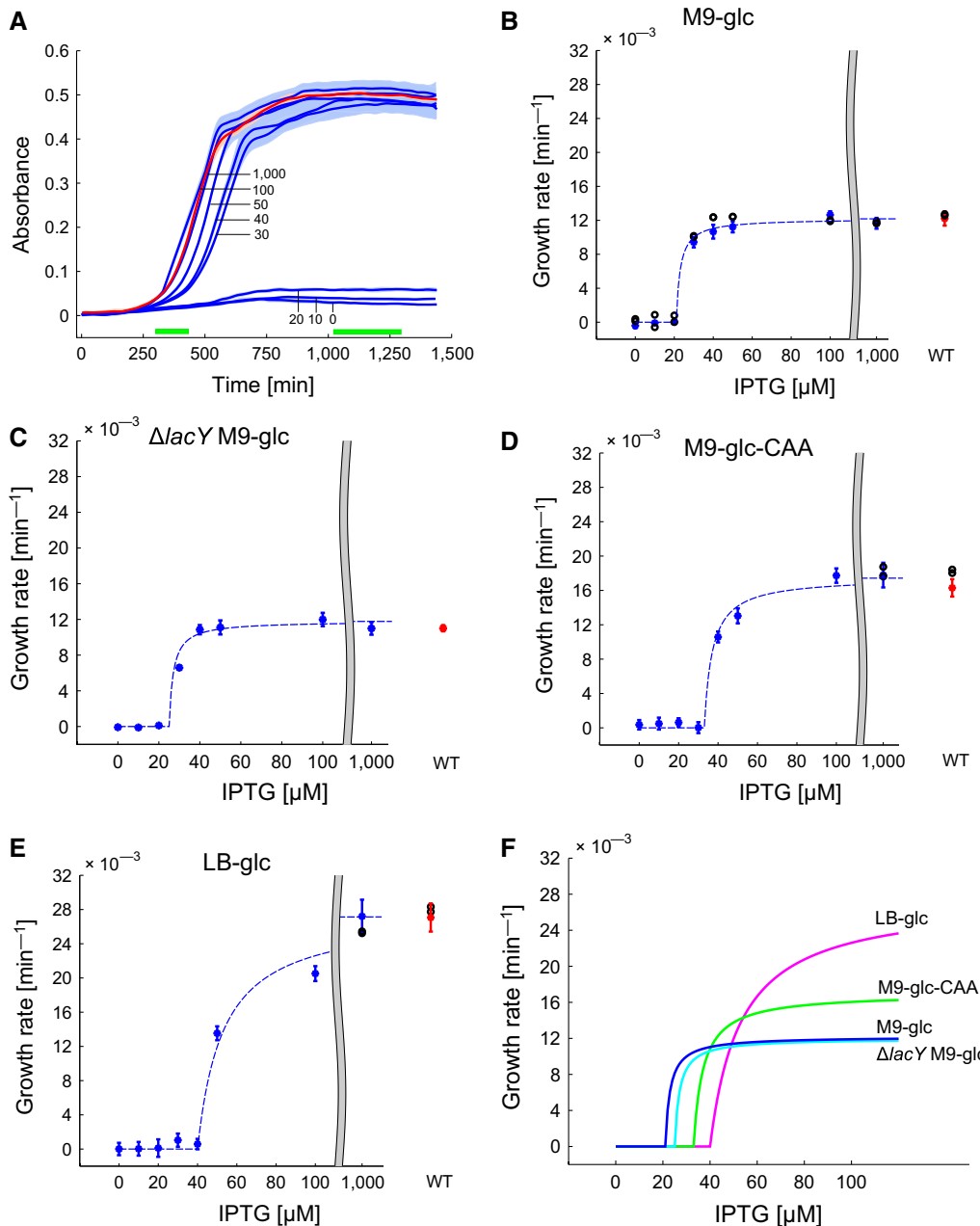

**Figure 2.  External control of the expression of *rpoBC* yields a growth switch in *E. coli*.**

A   The growth kinetics of the R strain changes as a function of the concentration of IPTG added to M9 minimal medium supplemented with 0.2% glucose. Blue curves are averaged absorbance measurements at 600 nm (five biological replicates), for increasing concentrations of IPTG (0, 10, 20, 30, 40, 50, 100, 1,000 μM). Shaded areas represent ± two standard errors of the mean. The red curve represents the growth kinetics of the W strain. The growth rates are typically computed in the time intervals indicated by the green bars.

B   The growth rate of the R strain responds in a switch-like manner to IPTG addition, with a threshold between 20 and 30 μM IPTG. The figure shows the growth rate estimated from the absorbance data, as explained in the main text, for both the R strain (blue) and the W strain (red). The reported growth rate is the mean over five replicates and the error bars are defined as ± two standard errors of the mean. The black circles represent the growth rates obtained with the R and W strains in shake flask experiments, which are in very good agreement with those obtained in microplate experiments. The dashed curve is obtained by fitting a Michaelis–Menten function to the data above the IPTG threshold and setting its value to 0 below the threshold.

C   Idem for M9 medium supplemented with 0.2% glucose in R-Δ*lacY* strain.

D   Idem for M9 minimal medium supplemented with 0.2% glucose and 0.2% casamino acids.

E   Idem for LB medium supplemented with 0.2% glucose.

F   Comparison of the growth-rate response of the R strain to different concentrations of IPTG. In all conditions, a switch occurs, in the sense that the growth rate reacts abruptly to a small increase in IPTG just above the threshold. The R-Δ*lacY* strain behaves quite similarly to the R strain in the reference medium (M9 medium with 0.2% glucose), indicating that the growth switch is not due to the cooperativity of the *lac* induction system.

Source data are available online for this figure.

                         

due to the positive feedback loop involved in inducer uptake. This apparent cooperativity can be strongly reduced by deleting the gene encoding the lactose transporter LacY. We therefore constructed the W-Δ*lacY* and R-Δ*lacY* strains to see whether the deletion of the lactose transporter gene affected the growth switch. As can be seen in Fig 2C, the engineered strain still functions as a growth switch and this phenotype is therefore not attributable to positive feedback in the inducer (IPTG) uptake.

Does the switch also work in other growth media? In order to test this, we performed the same experiment as above in minimal M9 medium supplemented with glucose and casamino acids and in rich LB medium supplemented with glucose, both supporting higher growth rates. The results are shown in Fig 2D and E. The maximum growth rates supported by these media are indeed higher ($0.016 \pm 0.0005$ min$^{-1}$, corresponding to a doubling time of 42 min, in the former, and $0.027 \pm 0.0008$ min$^{-1}$, corresponding to a doubling time of 27 min, in the latter). A similar growth switch is seen though, in the sense that as the inducer concentration crosses a threshold, the growth rate steeply increases from 0 to the maximum growth rate. Interestingly, as the medium becomes richer, supporting a higher maximum growth rate, the IPTG threshold at which growth starts increases as well (Fig 2F). Whereas for M9 minimal medium with glucose the switch occurs between 20 and 30 μM, supplementing this medium with casamino acids shifts the threshold to between 30 and 40 μM. In LB medium with glucose, between 40 and 50 μM of IPTG is needed for growth. The observed positive correlation between the maximum growth rate supported by a medium and the growth-switching threshold was confirmed by a fourth example, minimal M9 medium supplemented with succinate (Fig EV1). Notice that at higher growth rates proteins are diluted faster, which may require higher compensating levels of *rpoBC* expression.

In conclusion, the results of the growth experiments demonstrate a surprising switching behavior of the strain with engineered control of *rpoBC* expression, occurring in several different media supporting different maximal growth rates. The control experiment with the Δ*lacY* strain shows that the switching behavior does not depend on the cooperativity of the classical *lac* induction system, but is due to internal regulatory mechanisms coupling *rpoBC* expression to growth.

## Dependence of the growth rate on the concentration of RNA polymerase

While varying the IPTG concentration in the medium thus allows growth of the R strain to be switched on and off, the question can be asked whether growth-rate control is effectively achieved through variations in the concentrations of the β and β′ subunits of RNA polymerase.

In order to quantify the intracellular accumulation of ββ′, we constructed derivative strains of R and W carrying a translational fusion of the *rpoC* gene and the gene coding for the fluorescent protein mCherry, called "R-*rpoC*-mCherry" and "W-*rpoC*-mCherry", respectively (Appendix Fig S3). The fusion has been shown not to affect the activity of the tagged RNA polymerase (Bratton *et al*, 2011, and Fig EV1). Since the *rpoB* and *rpoC* genes are included in the same operon and no post-transcriptional regulatory mechanisms specific for the one or the other subunit are known, it is plausible that the quantity of tagged β′ is representative of the quantity of β as

well. This assumption is validated by recent ribosome profiling experiments showing that the synthesis rates of the β and β′ subunits (in numbers of molecules per generation in balanced growth) are the same within the limits of experimental error (Li *et al*, 2014).

Figure 3A shows the concentration of β′ once the bacteria have reached steady-state growth characteristic for a specific IPTG concentration, as explained in the previous section. The concentration of the fusion protein was determined from the measurements of the fluorescence signal and the absorbance at 600 nm, while correcting for the maturation time of the mCherry reporter (see Materials and Methods and Appendix Text S5 for data analysis procedures). As can be seen, a low IPTG concentration leads to a low β′ concentration in the R strain, most of which probably originates from the preculture. For IPTG concentrations above a threshold between 20 and 30 μM, however, the β′ concentration starts to increase. This implies that the higher growth rate reached for higher IPTG concentrations is associated with higher concentrations of β′, as expected if growth-rate control effectively depends on variations in the concentration of the *rpoBC* products. When plotting the growth rate as a function of the β′ concentration (Fig 3B), a saturation effect is observed. Above a certain concentration, an increase in β′ does not lead to a higher growth rate, indicating that RNA polymerase, or at least its β′ subunit, is no longer growth-limiting.

The plot in Fig 3B allows for two other interesting observations. First of all, the dependence of the growth rate on the β′ concentration appears to be highly switch-like or ultrasensitive, in the sense of Koshland *et al* (1982) [see Ferrell and Ha (2014) for a recent review]. A Hill function with a coefficient of 10 fits the data well, although more data may be needed to unambiguously establish a precise quantitative relation. The data indicate that growth requires a threshold level of RNA polymerase and the associated transcriptional activity, including transcription of metabolic enzymes, to switch from zero to maximal growth. A second observation is that the β′ concentration in the W strain lies just above this threshold level. This suggests that RNA polymerase levels in wild-type *E. coli* have been optimized in order to ensure maximal growth with minimal investment in the synthesis of these abundant and therefore costly proteins (Dykhuizen *et al*, 1987; Rest *et al*, 2013).

We performed the same experiment with the R-*rpoC*-mCherry and W-*rpoC*-mCherry strains in other growth media, in particular M9 medium supplemented with glucose and casamino acids and M9 medium supplemented with succinate, that is, well-defined media supporting a higher and a lower maximal growth rate, respectively, than M9 medium with glucose. The results are shown in Fig 3C and D and confirm the observations made for our reference conditions. We tried to determine the relation between β′ concentration and growth rate in LB as well, but did not succeed in obtaining a reliable steady-state concentration in this complex medium. Interestingly, a comparison of panels B–D of Fig 3 suggests that the dependence of the growth rate on the β′ concentration is remarkably similar across the different conditions. While the RNA polymerase concentration required for growth thus seems to be medium-independent, in richer media a higher activity of the *rpoBC* promoter is necessary to attain this concentration. This is consistent with Fig 2, which shows that the IPTG switching threshold shifts to a higher concentration in media supporting higher growth rates.

The results in Fig 3A–D confirm the motivating idea underlying the R strain, namely that for low concentrations of IPTG the

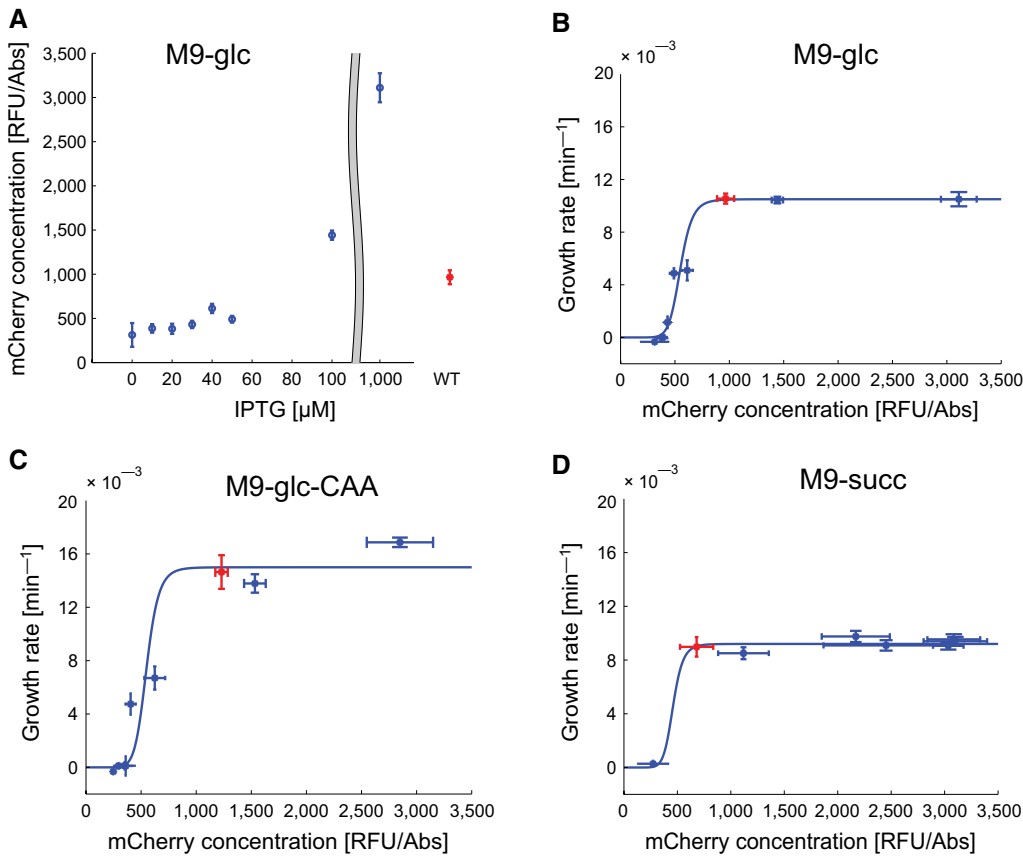

**Figure 3. External control of the expression of *rpoBC* modulates $\beta'$ availability.**

A   Quantification of the $\beta'$ subunit of RNA polymerase using a chromosomal fusion of the *rpoC* gene with the gene encoding the mCherry fluorescent reporter protein. We reconstruct the (relative) concentration of the $\beta'$ subunit from the fluorescence and absorbance measurements, as described in Materials and Methods and Appendix Text S5. The $\beta'$ levels are seen to be negligible for low concentrations of IPTG and strongly increase at higher concentrations of IPTG. The plot shows the mean of five replicates with error bars defined as $\pm$ two standard errors of the mean.

B   Quantitative dependence of the growth rate on $\beta'$ concentrations, computed from the data in (A) and the measured growth rates at the different IPTG concentrations. The color code is the same as in Fig 2B. The response curve is strongly sigmoidal, with wild-type levels of $\beta'$ situated just above the threshold. The blue curve is a Hill function with a Hill coefficient of 10.

C   Idem, for M9 minimal medium supplemented with 0.2% glucose and 0.2% casamino acids.

D   Idem, for M9 minimal medium supplemented with 0.2% succinate.

Source data are available online for this figure.

concentrations of $\beta$ and $\beta'$ become growth-limiting and cause growth arrest. As a further control, we quantified the relative availability of the $\beta'$ subunit with respect to $\alpha$, the other subunit of the RNA polymerase core complex, by Western blotting. More precisely, we quantified the fraction of total protein that is $\alpha$ and $\beta'$ in the R strain growing in minimal M9 medium with glucose for different concentrations of IPTG (Materials and Methods). We found that decreasing the IPTG concentration reduces the $\beta'/\alpha$ ratio (Fig EV2). The $\beta'$ subunit thus becomes limiting in the formation of the RNA polymerase core complex when *rpoBC* expression is lowered.

We also assessed global physiological changes in the cell by extracting and quantifying the total amounts of protein and RNA (Materials and Methods). The RNA/protein mass ratio was found to be approximately constant, at a value of around 0.15, in the W and R strains grown with 0 or 1,000 µM IPTG (Fig EV3). Interestingly, Scott *et al* (2010) obtained a similar constant RNA/protein mass ratio when reducing transcriptional activity in a different way, by

means of an antibiotic blocking transcription initiation (rifampicin). This constant ratio is characteristic of transcriptional inhibition, as translational inhibition with chloramphenicol was observed to increase the RNA/protein mass ratio (Scott *et al*, 2010). This observation provides further evidence that growth arrest in the absence of IPTG is indeed due to the repression of the *rpoBC* genes and the resulting lack of RNA polymerase available for transcription.

### Reversibility of the growth switch

Do growth-arrested cells recover a normal growth phenotype when RNA polymerase is provided again? This question is important for applications of the growth switch, where we would like to alternate periods of growth and growth arrest. The *rpoBC* genes are essential and the very low levels of RNA polymerase in R cells without IPTG, as observed in Fig 3A, may lead to cell death or to stress responses that ultimately hamper growth recovery. For example, exposure of

wild-type *E. coli* to rifampicin, even for a short period of 10 min, causes 95% of the cells to die (Sat *et al*, 2001).

In order to test the reversibility of growth arrest, we monitored the growth kinetics of the R strain at the single-cell level using a microfluidics device (Wang *et al*, 2010; Gasset-Rosa *et al*, 2014). The device traps cells in a channel about 1.5 times the diameter of a bacterial cell and about 20 times the length of an exponentially growing cell. The constant flow of medium assures invariant growth conditions and allows a rapid change of medium. In our case, the medium change consists in adding or removing IPTG from the minimal glucose medium. Growth of the bacteria was observed by time-lapse microscopy to obtain phase-contrast and fluorescent images (Fig 4 and Movie EV1). The growth rate of the individual R cells and their descendants was quantified by measuring the cell length of the newly formed bacteria in successive frames of the time-lapse microscopy movie (Appendix Fig S7).

R cells trapped in the dead-end channels were initially grown in the presence of 1,000 μM IPTG for 13 h in minimal glucose medium. The synthesis of RNA polymerase is strong and the cells normally grow and divide until they completely fill the dead-end channel (Fig 4). Newborn cells are then washed out into the large channel that supplies the medium. After 13 h of growth, IPTG was removed from the medium. An interesting observation is that in the absence of IPTG the cells no longer divide, but keep growing to a length of about 20 μm 6 h after IPTG removal (Fig 4). This increase in cell size was also observed when sampling R cells from a batch culture growing in the same conditions, and visualizing them by fluorescence and phase-contrast microscopy (Appendix Fig S8). Despite the increase in cell size, the growth rate of the cells continually and linearly decreases after an initial lag of about 1 h (Fig 4). After about 8 h without IPTG, in order to avoid elongated cells from being washed out of the wells by the medium flow on top of the device, we added IPTG back to the growth medium. 1.5 h after this new buffer change, probably the time required for RNA polymerase and other, unidentified cell components to reach a critical concentration, 90% of the 950 tracked cells resumed normal cell division and recovered the morphology and growth rate observed before growth arrest (Fig 4).

The results show that growth arrest, due to an externally effected decrease in the concentration of RNA polymerase, is a reversible process. In order to verify that this remains the case over several cycles of induction and repression of the *rpoBC* operon, we performed a two-day microplate experiment in which R cells were repeatedly switched between minimal glucose medium with high and low IPTG concentrations, respectively (Fig EV4). The growth kinetics over the four cycles turned out to be very reproducible, both in growth rate and in duration to recover a reference population density after redilution of the growth-limited cells into a

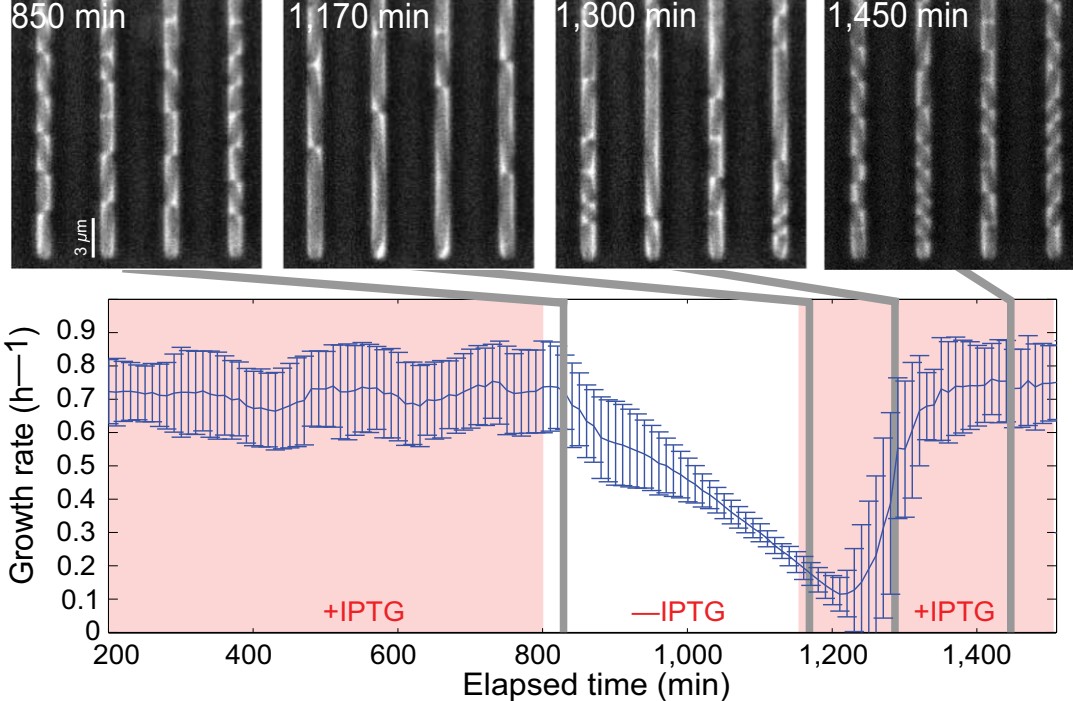

**Figure 4.  Growth arrest by external control of *rpoBC* is reversible.**
The R strain was grown in a microfluidics device and phase-contrast images were acquired every 10 min. The cells were grown in M9 minimal medium with glucose, initially in the presence of 1,000 μM IPTG. 6 h after removing IPTG from the medium, the cells are elongated. About 100 min after adding back 1,000 μM IPTG into the medium, the elongated cells divide and resume normal growth. See Movie EV1 for details of the growth kinetics. The growth rate of individual cells during each generation was computed as described in Appendix Fig S7. The growth rates in the plot are the (weighted) mean of the growth rates of 100 individual cells. The error bars correspond to ± one standard deviation.

Source data are available online for this figure.

medium containing a high concentration of IPTG. In addition, the cells rapidly transition to a high growth rate after the latter redilution. These observations show that growth arrest is reversible in a repeatable way and that the efficiency of recovery of growth-limited cells does not diminish after several cycles.

## Growth switch enables the improvement of metabolic production yields

An exciting application of the growth switch would be to use it as a tool for improving biotechnological processes. Arrest of the expression of *rpoBC* gives rise to slowly growing or non-growing cells with a functional metabolism, which opens up the perspective to utilize substrates supplied in the medium for the synthesis of specific metabolites of interest rather than biomass. Notice that contrary to classical biotechnological approaches, in which the fluxes in specific pathways are favored or disfavored by overexpressing or deleting enzymes (Bailey, 1991; Stephanopoulos & Vallino, 1991), limiting the activity of the gene expression machinery diminishes the demand for the building blocks of protein and RNA synthesis. The resources thus released can potentially be redirected toward pathways involved in the synthesis and secretion of a target product, whose enzymes present at the time of growth arrest remain functional.

Can our growth switch be exploited to implement the strategy outlined above? In order to provide a proof-of-principle of its interest as a biotechnological production platform, we have transformed the R strain into a glycerol production strain. We have measured if the growth-arrested cells of this strain are capable of converting glucose available in the growth medium to glycerol. Moreover, we have compared the yield of glycerol production in these growth-arrested cells with the yield obtained in an equivalent W strain.

We constructed a plasmid that expresses two yeast genes (*GPP2* and *GPD1*) under the control of a strong, constitutive promoter (Appendix Fig S4). The genes encode enzymes catalyzing the conversion of dihydroxyacetone phosphate to glycerol (Wang *et al*, 2001). In order to optimize glycerol production, we codon-optimized the yeast genes for *E. coli* and we translationally fused the two metabolic enzymes in order to maximize their metabolic activity [as shown previously, see Meynial-Salles *et al* (2007) and Liang *et al* (2011)]. Both the W strain and the R strain were transformed with this plasmid, giving rise to glycerol production strains labeled W-gly and R-gly, respectively. It is well known that the natural glycerol production and consumption pathway in *E. coli* is not active in the presence of glucose, due to carbon catabolite repression of the glycerol kinase enzyme (Deutscher *et al*, 2006). That is, wild-type *E. coli* cells do not produce glycerol when growing on glucose and, before glucose in the medium has been depleted, they do not consume glycerol when growing on a mixture of glucose and glycerol. We verified that this is also the case for the R strain (Appendix Fig S9).

The W-gly and R-gly strains were grown in shake flasks in minimal glucose medium without IPTG and samples were periodically removed to measure the optical density of the culture as well as the glucose and glycerol concentrations in the medium using coupled enzyme assays (Materials and Methods). Figure 5A and B shows the results. In the W-gly strain, glucose is exponentially consumed to

fuel growth while glycerol accumulates as a by-product, only to be taken up again once glucose is exhausted, at 500 min, allowing growth to continue at a lower rate. As expected, the growth rate of the R-gly strain is the same as in the W-gly strain in the beginning of the experiment, but strongly reduced after the initially high RNA polymerase concentration from the preculture has been diluted out, around 400 min. Notice that the R strain continues to produce glycerol when its growth slows down. More than half of the production has occurred by the end of the transition of exponential growth to growth arrest, around 700 min, but glycerol continues to be produced thereafter, thus demonstrating that the growth-arrested cells remain metabolically active. There is no evidence of glycerol consumption after glucose depletion, beyond 1,000 min. This is consistent with the conclusion that the concentration of RNA polymerase has fallen below a critical level supporting protein synthesis after growth arrest (Fig 3). Given that the glycerol production pathway in yeast is not reversible (Wang *et al*, 2001), the assimilation of glycerol after glucose depletion requires *de novo* synthesis of enzymes in the *E. coli* pathway, which is no longer possible at this stage.

But does the growth-arrested R-gly strain produce glycerol at a higher rate than the wild-type strain W-gly? We computed the time-varying yield, given by the ratio of the specific glucose uptake and glycerol production rates (Materials and Methods), in the time interval in which these rates are well defined, that is, for non-negligible glucose and glycerol concentrations (between 250 and 450 min for the W strain, and between 250 and 700 min for the R strain). As can be seen in Fig 5C and D, in the beginning the yields in the W-gly and R-gly strains are equal. As RNA polymerase starts to be diluted out and the growth rate decreases in the R-gly strain, the glycerol production yield doubles, and reaches its maximum at the end of the slow-down phase. On the contrary, the W-gly strain continues to grow at the same rate and produces glycerol at the same yield. This result is indicative of an increased allocation of resources to the production of glycerol at the depense of biomass synthesis in the R strain when stopping *rpoBC* expression. It is remarkable that the maximum glycerol production yield thus obtained is not far from the theoretical maximum yield (0.61), computed by means of a genomewide flux balance model (Feist *et al*, 2007; see Materials and Methods for details).

The results in Fig 5 provide a proof-of-principle of the interest of the growth switch for biotechnological applications. While a substantial improvement of the glycerol yield could be obtained by controlling the gene expression machinery, this does not mean that other approaches to growth arrest, such as treatment with antibiotics or the removal of an essential nutrient, could not produce equally high yields for this particular problem, the conversion of glucose to glycerol. We have tested our approach on the production of glycerol, a substrate listed as one of a dozen top value-added chemicals that can be produced from sugars via biological or chemical conversions (Werpy & Petersen, 2004), but the strategy has not been tailored in any way to glycerol production. While the general resource reallocation principle may also apply to other metabolites, optimal performance could require other well-known constraints for the engineering of metabolic pathways, such as cofactor imbalances (Auriol *et al*, 2011) or the accumulation of toxic intermediates (Kizer *et al*, 2008), to be specifically addressed in each case.

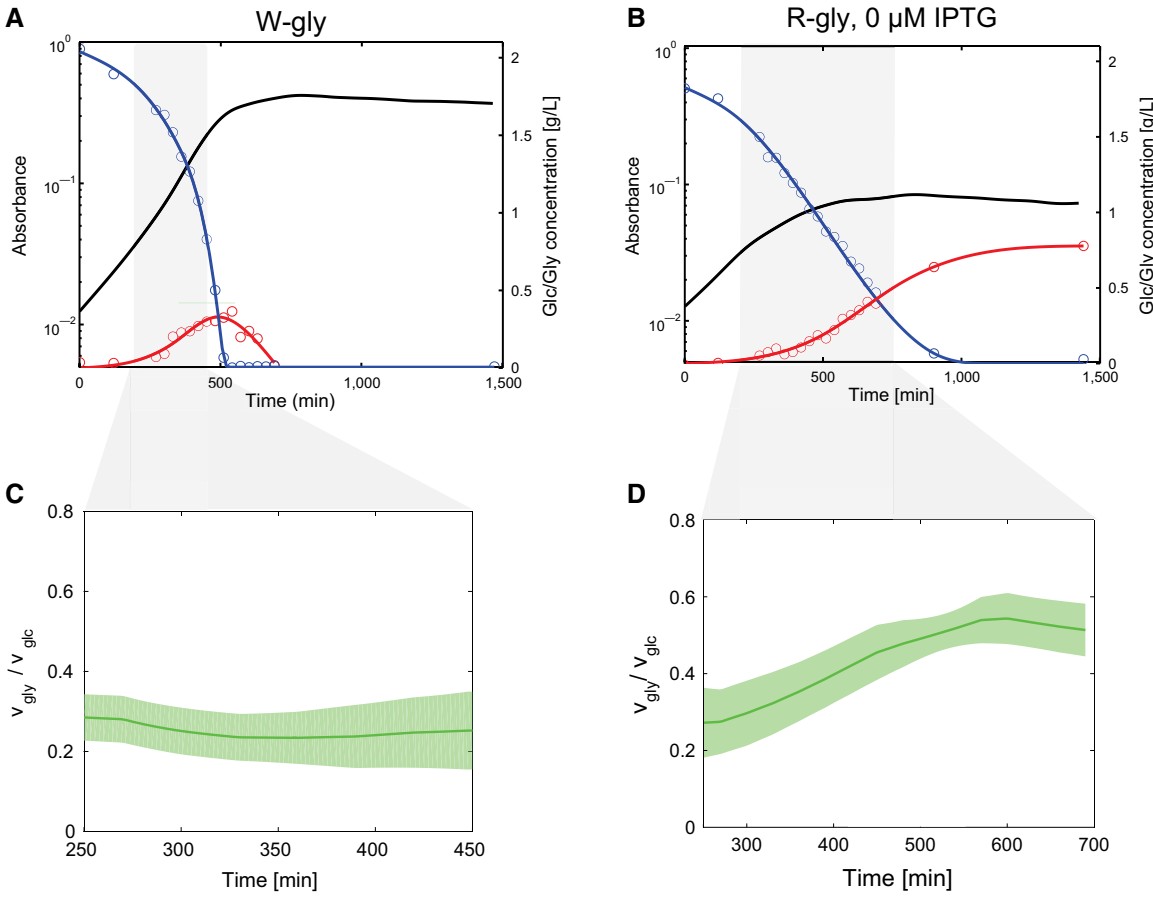

**Figure 5. Growth arrest by external control of *rpoBC* increases glycerol production yield.**

A   The glycerol-producing W strain (W-gly) was grown in shake flasks in M9 minimal medium supplemented with 2 g l⁻¹ glucose. The optical density (OD$_{600}$, black), the glycerol (red), and glucose (blue) concentrations were measured in samples taken at intervals of about 30 min using coupled enzyme assays (Materials and Methods). We show here the results of a typical experiment out of six biological and technical replicates.

B   The R-gly strain was grown in the same conditions as in (A). In the absence of IPTG, transcription of the *rpoBC* operon is inhibited, which causes RNA polymerase from the preculture to be diluted out, leading to growth arrest. Whereas the W-gly strain has consumed all of the available glucose around 500 min, R-gly continues to consume glucose and produce glycerol until around 1,000 min.

C   The instantaneous yield of glycerol production in the W-gly is computed by dividing the glucose consumption rate by the glycerol production rate, in a time interval in which the derivatives of the glucose and glycerol concentration curves are well defined (between 250 and 450 min).

D   Idem for the R-gly strain (in the time interval between 250 and 700 min). As can be seen, the yield in the W-gly strain remains constant over the time interval, whereas the yield in the R-gly strain doubles when RNA polymerase becomes growth-limiting. The curves show the mean of six experiments and a confidence interval equal to ± two standard errors of the mean.

Source data are available online for this figure.

## Stability of the growth switch

The phenotype imposed upon the bacteria when stopping *rpoBC* expression—growth arrest in conditions favorable for growth—is highly counterselected and thus potentially unstable. Genetic instability arises from mutations that disable the external control of *rpoBC* expression and allow growth even in the absence of IPTG. This phenomenon is similar to the appearance of rifampicin-resistant mutants in a bacterial culture, that is, of bacteria that grow despite the presence of this antibiotic. Such strains carry mutations in the coding region of *rpoB* (Jin & Gross, 1989 1998; Conrad *et al*, 2010). In our case, mutations could target LacI (disabling its capacity to bind DNA) or modify the promoter region (degrading LacI binding sites or transforming an upstream sequence into a new

promoter). Although loss of a counterselected phenotype is inevitable in the long run, we require our growth switch to outperform rifampicin and, in view of the envisaged biotechnological applications, to remain stable for at least one day.

Does our construction satisfy these criteria? As a first observation, we note that even after 24 h of growth in minimal M9 medium with glucose, in the absence of IPTG, no escape of the R strain is observed (Fig 2A). Some precautions were necessary to achieve this stability, such as a design comprising three copies of the *lacI* gene and a chromosomal instead of a plasmid-borne induction system (Fig 1). The stability of the growth-arrested R strain was confirmed in an additional experiment, where none of the 33 parallel cultures had resumed growth after 24 h and only two after 48 h (Fig 6A). This result should be compared with a similar experiment using the

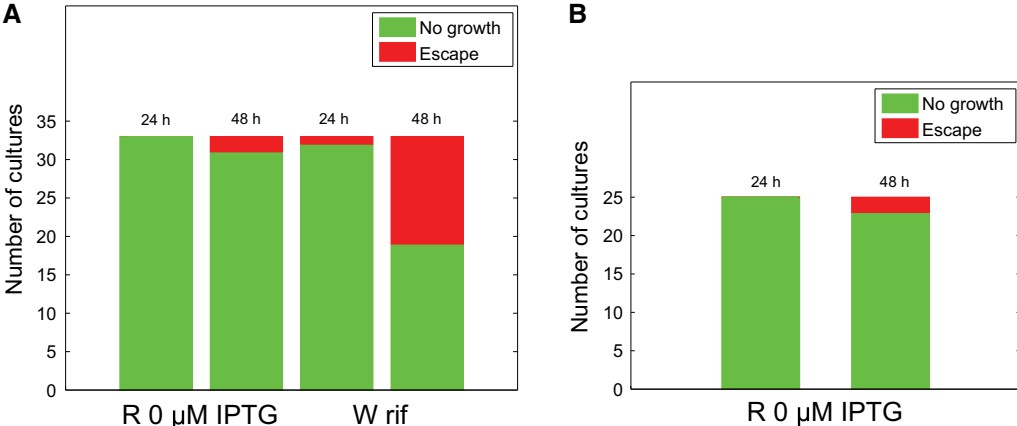

**Figure 6. Experimental test of genetic stability of growth switch.**

A The W strain was grown in a microplate as described in the Materials and Methods section, in M9 minimal medium with 0.2% glucose and 50 μM rifampicin. Idem, the R strain was grown in M9 minimal medium with 0.2% glucose and no IPTG. We used 33 replicate cultures (from 11 independent precultures started from different isolated clones on a Petri dish). The bar plot shows the number of cultures of the two strains that did not grow or had escaped, after 24 h and 48 h. Growth arrest in the R strain without IPTG is seen to be more stable than in the W strain treated with rifampicin (2 vs. 14 escapers after 24 h). Moreover, no cultures of the R strain without IPTG had escaped after 24 h.

B The R strain was grown in M9 minimal medium with 0.2% glucose and no IPTG in a shake flask containing 10 ml of growth medium. We used 25 independent replicate cultures, each started from a different isolated clone. The results are in agreement with the microplate experiment in (A) (no escapers after 24 h, 2 escapers after 48 h).

W strain, where growth was arrested by adding 50 μM rifampicin to the growth medium. After 48 h of growth, 14 of the 33 cultures had escaped (one after 24 h, see Fig 6A). The stability of our strain is thus superior to the wild-type strain treated with antibiotics. The absence of escapers after 24 h was confirmed when the experiment was repeated in the larger volume of shake flasks rather than in a microplate (Fig 6B).

The results also show that, when growth experiments last longer than 24 h, escape from growth arrest in cultures without IPTG eventually does occur in some cases. In order to test the nature of the escape in the R strain, we harvested the escaper cells in the above experiment and restarted the protocol of Fig 2 in minimal M9 medium with glucose and no IPTG from the preculture phase onwards. All harvested cultures immediately resumed growth (albeit often at a lower growth rate), suggesting that escape was due to mutants that were either present from the start or that occurred during the experiment, but not a consequence of phenotypic heterogeneity in an isogenic population, as observed in other contexts (e.g. Robert *et al*, 2010; Kotte *et al*, 2014; Solopova *et al*, 2014; van Heerden *et al*, 2014) The genetic stability of the strain could be further improved, for example, by putting the expression of another important subunit of RNA polymerase, the housekeeping sigma factor $\sigma^{70}$, under the control of an inducible promoter. Only the very unlikely event of two independent mutations would then produce an RNA polymerase holoenzyme functional in growth-arrest conditions.

# Discussion

The competition between different bacterial species is based on achieving optimal growth for a given environmental condition. As a consequence, growth rate is arguably the most important physiological property of bacteria. The ability to externally control the growth rate is not only instrumental for understanding how the cell optimizes its growth rate by allocating the available resources to gene expression, metabolism, and other cellular functions, but also holds promise for improving the yield and productivity of biotechnological processes. If it turns out to be possible to uncouple the capacity of the cell to produce certain metabolites of interest from its capacity to produce macromolecules for growth, incoming nutrient fluxes could be redirected toward specific metabolic pathways instead of feeding into biomass synthesis.

## Medium-independent and reversible growth switch

We developed a general-purpose growth switch for bacterial cells by putting the transcription of the large subunits of RNA polymerase, $\beta$ and $\beta'$, under the control of an IPTG-inducible promoter. It is well known that inactivation of the $\beta$ subunit of RNA polymerase, for example, by treating the cells with the antibiotic rifampicin, stops cellular growth (Campbell *et al*, 2001). However, the potential of a genetically engineered limitation of a central component of the gene expression machinery for growth-rate control and its biotechnological applications has not been exploited thus far.

Surprisingly, by changing the expression of the *rpoBC* genes via the concentration of the inducer IPTG, added to the growth medium, we can switch the growth rate of the R strain between zero and the maximum supported by a given medium (Fig 2B–E). This switching phenotype does not depend on the cooperativity of the *lac* induction system, since it is preserved in a strain from which the lactose transporter gene *lacY* has been deleted. Two other regulatory mechanisms that have been shown to contribute to the effective cooperativity of the IPTG response, transcription regulation by Crp·cAMP and DNA looping (Kuhlman *et al*, 2007), are also absent in the induction system used here. Instead, our data suggest that the growth switch

originates in the nonlinear dependence of growth on the RNA polymerase concentration, quantified by means of an mCherry tag on $\beta'$ (Fig 3A). Further work should quantify this ultrasensitive response more precisely and explain how it arises from the biochemical reaction networks and regulatory interactions coupling RNA polymerase concentrations to protein synthesis rates and metabolic fluxes.

Whereas overexpression of the *rpoBC* genes at high concentrations of IPTG leads to a significantly higher accumulation of the $\beta'$ subunit than in wild-type cells, growth does not increase beyond the rate of wild-type cells in the same medium (Fig 3A and B). This is consistent with previous observations of the effects of overexpressing the *rpoBC* genes (Bedwell & Nomura, 1986; Guzman & Jimenez-Sanchez, 1986). Growth rate does not increase even when all RNA polymerase subunits are overexpressed (Gummesson *et al*, 2009), because other cellular factors become growth-limiting. On the other hand, overexpression of the *rpoBC* operon does not incur a growth-impairing metabolic load either. Dekel & Alon (2005) observed such an effect for the gratuitous overexpression of the *lac* operon, but the measured decrease in the growth rate (4%) is probably too small to be visible in our data. Contrary to the overexpression of RNA polymerase, lowering its concentration below the wild-type level slows growth dramatically (Fig 3B–D). This asymmetry suggests that *E. coli* has evolved in such a way as to limit the synthesis of the $\beta$ and $\beta'$ subunits to the minimum required for maximal growth. In the words of Rest *et al* (2013), the expression of the $\beta\beta'$-encoding genes in *E. coli* seems to be perched "on the edge of a fitness cliff". The idea that the "natural" RNA polymerase concentration in the cell is probably the minimum required for sustaining maximal growth is also consistent with the rapid response to IPTG removal in the microfluidics experiment (Fig 4).

The possibility to switch growth on and off is obtained by genetic construction and does not require any modification of the nutrient composition of the medium. This increases the flexibility of the proposed approach and makes growth-rate control medium-independent, in the sense that it is able to function in a large variety of media, supporting different maximum growth rates. We demonstrated medium-independence by testing the functioning of the growth switch in four different media, with different maximum growth rates (Fig 2 and Fig EV1). The threshold concentration of inducer necessary for growth increases when the medium supports a higher growth rate. An intuitive explanation for this threshold shift is that a stronger induction of the *rpoBC* genes is required to compensate increased dilution of $\beta\beta'$ at the higher growth rate.

The initial response when the IPTG concentration crosses the threshold is the same in all media: a sudden increase in the growth rate. In richer media supporting a high growth rate, however, full induction requires IPTG to vary over a larger range (Fig 2F). This could be explained by saturation of the *rpoBC* transcription rate at higher concentrations of IPTG, or by an increasingly higher concentration of RNA polymerase required for growth in rich media. Figure 3 shows that the relation between $\beta'$ concentration and growth rate is remarkably similar in the different media considered, suggesting that the second explanation is unlikely. However, the precision of the data does not allow answering the question unambiguously.

Another interesting property of the construction is that the imposed growth-rate limitation is entirely reversible. Figure 4 shows that removing IPTG from the microfluidics channels causes a gradual decrease in the growth rate of the cells of the R strain and a return to the original value once IPTG is supplied again. While the decrease in growth rate upon lowering the concentration of RNA polymerase is almost instantaneous, there is a lag of about 1.5 h after the addition of IPTG before growth-limited cells resume normal growth. This lag period is probably necessary for replenishing the pool of RNA polymerase and other cellular components necessary for cellular growth. The microfluidics experiment also reveals that, during the slow-down of the growth rate, cell division stops and elongated cells appear that redivide upon the addition of IPTG. The cause of this filamentous morphology is currently unknown. It might involve the bacterial SOS response (Justice *et al*, 2008), but could also be a consequence of the decrease in concentration of a protein necessary for cell division when RNA polymerase is diluted out (and transcription of this factor stops).

## Growth-arrested cells remain metabolically active and can be transformed into microbial production platforms

The classical approach toward metabolic engineering consists in modifying specific components of metabolism, e.g. overexpressing heterologous enzymes or enzymes catalyzing a rate-limiting step in a natural pathway. Although quite a number of success stories exist, these approaches quickly reach their limits due to the robustness or "rigidity" of the underlying regulatory networks (Stephanopoulos & Vallino, 1991). The networks have evolved to counteract genetic and physiological perturbations, maintaining flux distributions geared toward growth. The motivation of the approach proposed here is the idea that, instead of interfering with the functioning of specific pathways, a fruitful alternative would be to directly act upon the global regulatory mechanisms of the cell to draw away resources from growth toward the production of metabolites of biotechnological interest.

The decoupling of cellular growth and biotechnological production has attracted much interest in metabolic engineering, notably for obtaining so-called quiescent cells, that is, cells that have stopped growing but remain metabolically active (Flickinger & Rouse, 1993; Sonderegger *et al*, 2005). A variety of solutions have been proposed to render cells quiescent. One approach consists in shifting a bacterial culture to starvation conditions, for example, by depriving the cells of nitrogen, phosphate, or other substrates necessary for growth (Matin *et al*, 1995; Sonderegger *et al*, 2005). A disadvantage of imposing nutrient starvation is that it has detrimental consequences on the production capacity of the cell. In particular, nitrogen starvation is not possible when the target compound itself contains nitrogen atoms (amino acids, polyamides, …), while phosphate starvation perturbs the energy household of the cell. Other approaches specifically target the molecular machinery responsible for growth. For example, the overexpression of a small RNA regulating cell division has been shown to provoke cell-cycle arrest and enable a substantial overproduction of a heterologous protein, at least over a limited time interval (Rowe & Summers, 1999). This approach causes high cell mortality though, like the use of antibiotics, and more generally has not been designed for supporting alternated phases of growth and growth arrest.

The approach proposed in this manuscript aims at relieving the above problems by proposing an externally controlled growth

switch that is reversible and medium-independent. It does not require the cells to be maintained in starvation conditions and works in a variety of different media. Moreover, it allows periods of growth arrest to be alternated with periods of normal growth. The application of this growth switch has made it possible to obtain an important increase in the glycerol production yield, close to the theoretical maximum, in a growth-arrested *E. coli* strain (Fig 5). This has been achieved by simply transforming the R strain with a plasmid carrying an enzyme fusion endowing *E. coli* cells with the capability to produce glycerol while growing on glucose, without any further optimization, such as the deletion of the gene encoding triose phosphate isomerase, the enzyme intraconverting dihydroxyacetone phosphate (DHAP) and glyceraldehyde 3-phosphate (G3P) (Wang *et al*, 2001). This proof-of-principle shows the potential of the growth switch as a generic tool for metabolic engineering. However, a more comprehensive validation of the approach, by applying the growth switch to the production of other metabolites and comparing the outcomes with alternative growth-arrest strategies, is necessary to map its strengths and limitations.

The performance and range of applicability of the growth switch in biotechnological applications could be further improved by tailoring or fine-tuning specific regulatory networks of the cell. For example, the synthesis of enzymes necessary for the production of a metabolite of interest could be placed under the control of a heterologous RNA polymerase that is not subject to reduced expression in growth-arrested R cells, such as the RNA polymerase of bacteriophage T7 (Wagner *et al*, 2008). This would allow a larger share of the remaining resources for protein synthesis to be specifically oriented toward the synthesis of enzymes in the desired production pathway. In a similar way, introducing modifications in RNA polymerase sigma factors, either in a directed way or through random mutagenesis, could be used to preferentially redirect RNA polymerases to genes of interest (Alper & Stephanopoulos, 2007). Another interesting direction is the exploration of different temporal *rpoBC* induction patterns and their effect on product yield or other performance criteria, such as productivity. While our working hypothesis has been to aim at driving the system to a prolonged state of growth arrest, different strategies like sequences of exponential growth and growth slow-down could turn out to be more effective. The data in Fig 5 show that the optimal glycerol production yield is already attained before complete growth arrest.

### Growth switch is modular: toward interfacing the growth switch with physiological signals

The growth switch presented in this manuscript is controlled by external addition of an inducer to the medium. In the terminology of mathematical control theory (Isidori, 1995), the strain is an "open-loop controller". However, since the controller is a genetic element, the transcription of the *rpoBC* operon can be connected to other gene circuits and a variety of internal physiological signals. This would, for example, allow growth to be switched on and off depending on enzyme or metabolite concentrations in the cell, instead of alternating *rpoBC* expression in a predefined manner. Moreover, by adding a growth-rate sensor in the cell, growth could be stabilized at intermediate levels by enhancing natural feedback with additional synthetic feedback loops, implemented either in the bacterial cell (Dahl *et al*, 2013; Xu *et al*, 2014; Venayak *et al*, 2015) or in an external controller device (Milias-Argeitis *et al*, 2011; Uhlendorf *et al*, 2012; Menolascina *et al*, 2014). In other words, this opens the possibility to transform the strain into a "closed-loop controller", with exciting perspectives at the interface of control engineering and molecular biology.

The modularity of the growth controller, together with its previously mentioned medium-independence and reversibility, makes it a promising tool for fundamental studies of the global cell physiology of *E. coli* and potential applications in synthetic biology and biotechnology. Moreover, while all results reported in this manuscript have concerned *E. coli*, a similar method for growth-rate control can be implemented in other microorganisms with a single RNA polymerase.

## Materials and Methods

### Bacterial strains and plasmids

The strains are derivatives of the *E. coli* K-12 BW25113 strain (Baba *et al*, 2006). Our reference strain (W) carries two copies of the *lacI* gene under the control of a strong constitutive promoter (Appendix Fig S2) to provide a large amount of Lac repressor and to prevent the appearance of fixed mutations. In addition to these extra copies of *lacI*, our modified strain (R) carries an IPTG-inducible promoter driving the expression of the *rpoBC* operon (Appendix Fig S1). We also constructed a variant of the W and R strains with a markerless deletion of the *lacY* gene (W-Δ*lacY* and R-Δ*lacY*). We used bacteriophage P1 transduction of the kanamycin deletion cassette obtained from the BW25113 Δ*lacY* donor strain (Baba *et al*, 2006), and then eliminated the cassette by thermosensitive induction of the FLP recombinase (Zhang *et al*, 1998). The deletion of the lactose transporter LacY decreases the cooperativity of the response of the IPTG-inducible promoter (Kuhlman *et al*, 2007). The W-*rpoC*-mCherry and R-*rpoC*-mCherry strains are derivatives of the W and R strains, respectively, carrying a chromosomal fusion of the *rpoC* gene with the gene encoding the mCherry fluorescent reporter protein (Appendix Fig S3). The tagged β' subunit encoded by this construction allows the *in vivo* quantification of RNA polymerase (Bratton *et al*, 2011). We also transformed the W and R strains with a pUC57 plasmid expressing a fusion of the yeast genes *GPD1* and *GPP2* (Appendix Fig S4). The resulting fusion protein efficiently catalyzes the conversion of dihydroxyacetone phosphate to glycerol (Liang *et al*, 2011). These engineered glycerol production strains are labeled W-gly and R-gly, respectively. All strains and plasmids were verified by sequencing (Appendix Table S1).

### Growth conditions and growth-rate measurements

For all measurements, a single colony was picked from an LB IPTG (1,000 $\mu$M) agar plate and incubated overnight at 37°C in minimal M9 medium supplemented with 0.2% glucose (w/v) and 1,000 $\mu$M IPTG. The growth kinetics were measured either in a covered microplate or in a shake flask. At time zero, IPTG was removed from the overnight culture by centrifugation for 5 min at 14,000 $g$ and the cells were washed with fresh M9 medium without IPTG. This operation was repeated twice and the inoculum size was adjusted to the same $OD_{600}$ for all cultures. At the start of the experiment, the

cultures thus obtained were diluted into fresh medium (M9-0.2% glucose, M9-0.2% glucose supplemented with 0.2% casamino acid, M9-0.2% succinate, LB-0.2% glucose, or M9-0.2% glucose supplemented with rifampicin at a final concentration of 50 μM) with or without IPTG to an initial $OD_{600}$ of 0.01 or 0.001. Each well contains 150 μl of adjusted culture and a sterile 2-mm glass bead to improve aeration. The absorbance (600 nm) and the fluorescence (560/635 nm) were read every 1–2 min in an automated plate reader (Infinite 200 PRO series, Tecan), thermostated at 37°C. Each readout of the microplate was preceded by a 20-s stirring step (5 mm shaking amplitude). For the microplates used, absorbance values can be converted to $OD_{600}$ values by multiplication with a factor 2.34. In the time window of interest, in particular after RNA polymerase from the precultures has been diluted out (see the main text), the resulting growth curves were fitted with an exponential function to compute the growth rate. We computed the mean values of five biological replicates and their confidence intervals, consisting of ± two standard errors of the mean, which corresponds to 95%-confidence intervals in the case of a Gaussian distribution. We eliminated data from wells showing edge effects, in the rare instances where this occurred. The shake flask experiments were carried out in exactly the same way as above, except that samples were manually taken at regular time intervals (ranging from once per 30 min in M9 to once per 10 min in M9 with casamino acids and LB), and $OD_{600}$ values measured in a spectrometer. The shake flasks contained 40 ml of growth medium for a total volume of 200 ml. In the shake flask experiment used for determining the genetic stability of the strain, the flasks contained 10 ml of medium and the optical density was measured less frequently, with data points centered around 24 and 48 h.

### Quantification of $β'$ subunit of RNA polymerase using fluorescent fusion protein

The accumulation of the $β'$ subunit of RNA polymerase in W and R cells was quantified using the W-*rpoC*-mCherry and R-*rpoC*-mCherry strains in growth experiments carried out in the microplate, as described above. In order to compute an estimate of the mCherry concentration (in relative units) from background-corrected fluorescence and absorbance values, we applied a variant of the measurement models and data analysis procedures described in de Jong *et al* (2010), taking into account the maturation kinetics of the mCherry reporter. The details are given in Appendix Text S5.

### Quantification of RNA polymerase subunits using Western blotting

After 4–5 h of growth in M9 minimal medium supplemented with 0.2% glucose, cells were harvested by centrifugation at 10,000 *g* for 10 min at 4°C. The cells were stored at −80°C and lysed using the BugBuster Master Mix solution (Novagen) supplemented by a complete mini protease inhibitor cocktail (Roche) for 20 min at room temperature. The soluble cell extracts were collected by centrifugation at 14,000 *g* for 20 min at 4°C and the amount of total protein was measured using a Bradford assay (Biorad). In order to detect the α and $β'$ subunits of RNA polymerase by Western blotting, 25–35 μg of total protein were separated on a 4–12% NuPAGE gel (Invitrogen), transferred to Hybond-P membranes (Amersham),

blocked with non-fat dried milk, and incubated with 1/20,000 mouse anti-α (sc-101597 4RA2) and 1/5,000 anti-$β'$ (sc-101613 NT73) for 1 h at room temperature. Membranes were washed and incubated for another hour with secondary anti-mouse antibody conjugated with peroxidase. The Amersham ECL chemiluminescent kit was used to detect and quantify the signals. For the quantification of the Western blots, we acquired the images of the membranes using an amplified CCD camera. The digital Western blot images (Fig EV2) were analyzed using ImageJ (http://rsb-web.nih.gov/ij/). The density of each band was measured, and a constant background was subtracted based on the intensity of the surrounding area. The band intensity for the $β'$ subunit was divided by the quantified intensity of the α subunit. This ratio quantifies the relative expression value of $β'$ with respect to α in each strain at different IPTG concentrations.

### Quantification of total protein and RNA

W and R cells growing in a shake flask in M9 minimal medium supplemented with 0.2% glucose and 1,000 μM IPTG were sampled in mid-exponential phase ($OD_{600} = 0.3$), while R cells grown in the same conditions without IPTG were sampled after bacterial growth had ceased. Soluble protein extracts were prepared and quantified as described above. RNA was extracted using the RNeasy Mini Kit supplemented with the RNAprotect Bacteria Reagent (Qiagen) and quantified by spectrophotometry at 280 nm.

### Microscopy imaging and quantification of cell size

W and R cells were sampled after 5 h of growth in M9 minimal medium with 0.2% glucose in a shake flask. The cells were fixed with a 10% formaldehyde solution at room temperature for 20 min, pelleted by centrifugation at 10,000 *g* for 2 min, and resuspended in absolute ethanol. An aqueous solution of Hoechst dye was added to a final concentration of 10 μg ml$^{-1}$. For the quantification of cell size, between 1,200 cells for the R strain growing in 0 μM IPTG and 8,000 cells for the R strain growing in 1,000 μM IPTG were examined using an IX2 Olympus microscope and their cell area was calculated using the Scan$^©$ software of the manufacturer. See Appendix Fig S8 for further details.

### Microfluidics experiments

For the microfluidics assays, cultures were grown overnight at 37°C in M9 supplemented with 0.2% glucose and 1,000 μM IPTG. The overnight culture was diluted 50-fold into 10 ml of fresh medium at 37°C. When the culture reached $OD_{600}$ equal to 0.15, the cells were concentrated by centrifugation and injected into a microfluidics device (Wang *et al*, 2010; Gasset-Rosa *et al*, 2014). The cells were loaded by centrifugation of the device until more than 80% of the channels were filled with bacteria. We then passed fresh M9 medium supplemented with 1,000 μM IPTG through the device and started image acquisition for about 2 days. After 13 h, the syringe containing the medium was switched to the same medium without IPTG. 6 h later, the medium was switched back to 1,000 μM IPTG. Pictures of 40 fields were taken every 10 min, representing a total of about 1,000 individual channels. For details of the image processing method and the computation of the growth rate of individual cells

from the imaging data, see Appendix Text S6 and Appendix Fig S7, respectively.

## Quantification of extracellular glucose and glycerol concentrations

The external glucose and glycerol concentrations were measured using the commercially available D-glucose HK and Glycerol Assay Kits, respectively (Megazyme, Ireland). Sampling was performed during the microplate growth experiments described above. An appropriate well (150 µl) was sacrificed at regular time intervals, the culture volume centrifuged at 14,000 *g*, and the supernatant stored at −20°C. The glucose and glycerol concentrations were determined from absorbance measurements at 340 nm in a microplate reader (Infinite 200 PRO series, Tecan Group Ltd., Germany), by means of a calibration curve and appropriate dilutions, following the manufacturer's instructions.

### Computation of theoretical and actual glycerol production yields

The time-varying glycerol production yield $Y(t)$ is defined as the positive ratio of the specific glucose consumption rate and the specific glycerol production rate. This ratio can be computed from the time-derivative of the quantity of glucose and glycerol in the medium ($x_{glc}(t)$ and $x_{gly}(t)$ [g l$^{-1}$]) as follows:

$$Y(t) = -\frac{\mathrm{d}}{\mathrm{d}t}x_{gly}(t) \Big/ \frac{\mathrm{d}}{\mathrm{d}t}x_{glc}(t).$$

The concentrations of glucose and glycerol in the growth medium were measured as described in the previous paragraph (one replicate is shown in Fig 5). A spline was fit to the data, and the analytical derivative of the spline fit was used to compute the growth yield $Y(t)$. The spline derivative can be calculated with reasonable confidence only in the stages of the experiment where the concentrations of glucose and glycerol change significantly (shaded areas in Fig 5). The growth yields obtained from six experiments were averaged and a standard error of the mean calculated from these data (shaded curves in Fig 5C and D).

The theoretical production yield of glycerol has been computed using the genome-scale reconstruction of *E. coli* metabolism iAF1260-flux2 (Feist *et al*, 2007). Simulations were performed using the COBRAv2 Toolbox with GLPK as the linear programming solver (Schellenberger *et al*, 2011). The oxygen uptake flux was not constrained. In addition, we set the lower and upper bounds of the non-growth-associated maintenance flux to its default value (6.75 mmol gDW$^{-1}$ h$^{-1}$), the growth-associated maintenance flux to 0 mmol gDW$^{-1}$ h$^{-1}$, and the exchange flux of glucose to −1 mmol gDW$^{-1}$ h$^{-1}$. Production of glycerol was optimized and the yield determined by dividing the glycerol production flux by the glucose uptake flux (in units g gDW$^{-1}$ h$^{-1}$).

**Expanded View** for this article is available online.

## Acknowledgements

This work was partially supported by the Agence Nationale de la Recherche under project GeMCo (ANR-2010-BLAN-0201-02), the INRIA/INSERM Action d'envergure ColAge, the Investissements d'Avenir Bio-informatique programme under project RESET (ANR-11-BINF-0005), Labex grant Sorbonne Paris Cité, and Axa Foundation Chair. The authors thank Guillaume Baptist for help with microscopy imaging, Pierre Pautré for strain construction, Michel Page and Valentin Zulkower for help with data analysis, and Corinne Pinel and Ludowic Lancelot for laboratory assistance.

## Author contributions

JI, CDCGB, DR, SL, ABL, JG, and HdJ designed the research. JI, CDCGB, DR, JG, and HdJ developed constructions and/or performed the experiments. JI, CDCGB, DR, SL, JG, and HdJ analyzed the data. XS and YY contributed analytic tools. JI, CDCGB, DR, JG, and HdJ wrote the paper.

## Conflict of interest

The authors declare that they have no conflict of interest.

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
