## [Review Process File · Molecular Systems Biology]

A synthetic growth switch based on controlled expression of RNA polymerase

J erome Izard, Cindy D.C. Gomez Balderas, Delphine Ropers, Stephan Lacour, Xiaohu Song, Yifan Yang, Ariel B. Lindner, Johannes Geiselmann, Hidde de Jong

Corresponding author: Hidde de Jong, INRIA

Review timeline:	First Submission:	19 November 2013
	Editorial Decision:	11 December 2013
	Second Submission:	03 July 2015
	Editorial Decision:	10 August 2015
	Revision received:	09 September 2015
	Editorial Decision:	09 October 2015
	Revision received:	15 October 2015
	Accepted:	16 October 2015

Editor: Maria Polychronidou

Transaction Report:

1st Editorial Decision

11 December 2013

Thank you again for submitting your work to Molecular Systems Biology. We have now heard back from the three referees whom we asked to evaluate your manuscript. As you will see from the reports below, the referees acknowledge that the presented approach is potentially interesting for biotechnological applications. However, they raise substantial concerns on your work, which, I am afraid to say, preclude its publication in Molecular Systems Biology.

The reviewers raise significant concerns regarding the interpretation of the main findings and the conclusiveness of the study. In particular, they are not convinced that the presented experimental data provide sufficient support for the quantitative dependence of growth rate on RNA polymerase levels and they point out that in absence of a concrete example demonstrating the applicability and efficacy of the strategy, its relevance for biotechnological applications remains unclear. Moreover, they think that further experimental evidence is required to convincingly exclude the induction of stress responses and that several additional control experiments should be performed throughout the study.

Considering these rather substantial concerns, we feel that we have no choice but to return this manuscript with the message that we cannot offer to publish it.

Nevertheless, as the reviewers did have positive words for the goals and the potential relevance of the study for biotechnology, we would like to indicate that we would be willing to reconsider a new submission based on this work. Any new submission would need to include new experimental data rigorously addressing the concerns raised by the reviewers.

A resubmitted work would have a new number and receipt date. We recognize that this would involve substantial additional experimentation and analysis and, as you probably understand, we can give no guarantee about its eventual acceptability. If you do decide to follow this course then it would be helpful to enclose with your re-submission an account of how the work has been altered in response to the points raised in the present review.

I am sorry that the review of your work did not result in a more favorable outcome on this occasion, but I hope that you will not be discouraged from sending your work to Molecular Systems Biology in the future. In any case, thank you for the opportunity to examine this work.

REFEREE REPORTS

Reviewer #1:

In this work the authors propose a way for externally controlling the cellular growth rate of *E. coli*. This is achieved by replacing the promoter for the $\beta\beta'$ RNA polymerase subunits (encoded by *rpoB* and *rpoC*) by an inducible promoter, based on the *lac* system, and responsive to the synthetic compound IPTG. This allows the authors to achieve growth-rates for *E. coli* spanning over two-fold in range, without changing the composition of the growth media. The authors suggest this method for growth-rate modulation irrespective of media composition may serve as a valuable tool for studying bacterial physiology and advocate its utility for biotechnological applications.

The ability to externally control the growth rate of microorganisms is indeed a highly desirable trait in both the study of cellular physiology and in biotechnology, and as such this work poses an ambitious and important goal. Also, achieving this goal by controlling the concentrations of RNA polymerase is an interesting avenue of research, as this may alter the growth rate without disrupting the metabolic capabilities of the cells, which is crucial for the utility of this approach for biotechnology. However, I was not convinced that the experimental procedures undertaken by the authors to address this challenge are adequate for several reasons, as detailed below, and therefore highly doubt that the constructed strain may actually serve as a tool for both basic research and biotechnology.

First, the authors chose to modulate $\beta\beta'$ levels by an IPTG-inducible promoter, similar to the well-studied endogenous *lac* promoter. Being a model system for transcriptional regulation, much work has been dedicated to characterizing the induction curve of the *lac* promoter in response to external IPTG. Much of this work showed that *lac* expression in the population behaves like a switch- cells are either fully ON or fully OFF. As such, intermediate levels of IPTG do not induce intermediate levels of induction in all cells, but rather change the fraction of cells that are ON (see Novick and Weiner 1957, Ozbudak et al. 2004, Vilar et al. 2003, Robert et al. 2010 and references within). Notably, Izard et al. also observe this phenomenon in their synthetic construct when treating the cells with intermediate levels of IPTG (30 μ M) and observing that the population divides into two subpopulations displaying vary different morphology and growth rate (Supplementary figure 5). As such, it appears to me that the main goal of the authors, which is to modulate growth rate by varying levels of RNA polymerase for individual cells, has not been achieved by this methodology. Personally, I would be very apprehensive about any conclusion about growth rate and physiology which is based on this system, which generates mixed populations of growing and non-growing cells.

Second, while biotechnological applications are stated to be a strong motivation for this study, there is no experiment showing the biotechnological efficacy of this technique, not even as a proof of concept. I think this proof of concept experiment is important as a priori the experimental system appears to have several inherent limitations, which may inhibit its use in biotechnological settings. For example, the authors suggest that this system will enable to maintain slow growth rates, even in nutrient-rich environments, which is important for sustaining high metabolic rates. This is a very unstable situation, as any cell that due to mutation is able to circumvent the repression and grow faster, will quickly overtake the population and the advantage of the system will be lost. The authors provide a partial response to this issue by inserting to the genome two additional copies of the repressor *lacI*. Whereas this indeed reduces the probability of escaping repression by mutating *lacI*, it does not deal with other escape options such as mutating the *lacO* sites, duplicating the entire

genomic region etc. I was not convinced that this system may indeed prove stable in the long-term growth periods required for biotechnology.

Altogether, I think that the goal of experimentally modulating cell growth by changing RNA polymerase levels in cells is interesting. It would be very interesting to determine the quantitative dependence of growth rate on RNA polymerase levels. However, it appears that the genetic construct presented in this manuscript does not allow to properly address this issue. This construct does provide other interesting results, such as the morphology of the cells when RNA polymerase is repressed, yet these may be better presented in a different context asking different questions than those currently posed in the introduction.

Additional comments:

1. The methods do not discuss how μ_{\max} is calculated from the growth curves. Accordingly, it is not shown that the cells display exponential growth for all conditions. This is of importance because the authors pre-grow their strains in a media fully induced with IPTG. Therefore, if not calculated carefully, μ_{\max} may actually represent the residual growth with the RNA-polymerase left in the cells from the previous culture conditions.
2. The authors claim that their methods can span a large range of growth rates, from (almost) zero to the maximal growth rate possible in the tested environmental conditions. However, in figure 1C,D: it appears that with concentrations $< 30\mu\text{M}$ the strain does not exhibit balanced exponential growth. From $30\mu\text{M}$ to $1000\mu\text{M}$ the cells span only two-fold in growth rate. In most experiments in which growth rate is modulated by media composition, a span of at least four-fold is easily achievable. A span of two-fold, even if it was generated by a homogenous population, is a rather small range if one wants to study bacterial physiology.
3. The authors show how growth rate is dependent on IPTG concentrations (figure 2) and several points showing the dependence of β' abundance on IPTG. From a basic science perspective, it would be very interesting to characterize the dependence of growth rate on β' abundance. Is this relationship linear? (See Dykhuizen et al., 1987 for a similar analysis done for the lac genes or Rest et al., 2012 for this analysis on a yeast gene).
4. The analysis of gene expression seems rather superficial and not convincing. The authors should report the number of differentially-expressed genes and display the results also in a common graphical representation (e.g. heat map or scatter plot). Also, there are many issues with analyzing genome-wide expression datasets when global changes (such as changes in total RNA content) are at play, as presumably is the case in this experiment (Loven et al., 2012). There are ways to address these issues experimentally (Loven et al., 2012; Bakel & Holstege, 2008; van de Peppel et al, 2003; Islam et al, 2011), however it does not appear that any such controls were performed by the authors.
5. In the introduction there are some strong statements not supported by data. The early Copenhagen works showed that the physiological properties of the cell go together with growth rate. The authors state that these properties are determined by growth rate, which implies a causality that is highly debatable in the field (see Levy and Barkai, 2009 for evidence that the causality may be reversed). Accordingly, the authors declare that these properties are optimized, again a strong claim which is not claimed in the original papers and demands proof.

Reviewer #2:

The authors described the reversible control of cell growth rate by replacing the E. coli native RNA polymerase rpoBC promoter with an IPTG-inducible promoter. By tuning the concentration of the inducer IPTG, the authors successfully tuned the E. coli growth rate between zero and the maximal achievable growth rate. The authors further demonstrated that this growth-arrest state could be reversed by external supplementation of IPTG. In the absence of IPTG, the growth-arrest cells continue maintaining metabolic activity but undergoing no cell division -- an observation that is related with abnormal cell morphology and elongated cell shape. While these findings potentially could be important to optimal growth-rate control in synthetic biology and biotechnological applications, the scientific facts and the experimental data present in this study only weakly or indirectly support their major conclusion.

Specific comments:

- (1) The authors described that the intracellular concentration of the β and β' subunits relative to the α

subunit were determined (page 5). However, only the result of β' subunit was displayed (Fig. 2). The genes encoding β and β' subunits were located in an operon. Generally speaking, the gene proximal to the promoter in an operon transcribes more RNA than those behind it. But, translation may change protein contents in cells. So, the authors should give an explanation why the amount of β' subunit sets the overall concentration of RNA polymerase but not β subunits. Otherwise, the relative concentration of β subunits should be determined.

(2) As expected, the relative expression of the beta' protein is decreasing when the authors reduced the amount of IPTG in the culture (Figure 2). However, the expression of the alpha subunit is also subject to the control of the functional RNA polymerase. The expression of the reference protein (alpha subunit) appears remained constant when the level of externally added IPTG changes. How could this happen?

(3) It is unclear how the growth rate of the wild type strain would change at different concentration of IPTG. The authors demonstrated that the specific growth rate remained constant for the wild type strain in the absence (0mM) or the presence (1mM) of IPTG in Figure 1D. Chemical inducers, however, are generally known as stress elicitors that, in some cases, could trigger global transcription changes that are detrimental to cell growth. The authors are suggested to provide more data to support their conclusion.

(3) The growth kinetics obtained in microplates (Figure 1C) and microfluidic channels (Figure 4) are not enough to draw all the conclusions. Due to edge effects and transport limitations, cell growth rate estimated with these devices are generally considered as not accurate as compared with flask or chemostat experiments. The growth curves were somewhat different in microplate and flask for R strain and W strain under 1 mM IPTG (Figure 1C and Figure S3). It is safe to test the growth of R strain under all IPTG concentrations (as shown in Figure 1C) in flask instead of only 0 μ M, 30 μ M and 1mM.

(4) Cell growth rate are generally related with the amount of ribosomal proteins and the availability of cellular resources (such as amino acids) inside the cell. In order to refine this article, the authors are suggested to quantify the amount of ribosomal proteins and ribosomal RNAs (rRNAs) at different level of IPTG.

(5) "It is reassuring that the chosen method for growth limitation does not trigger a major stress response or other profound physiological modifications" (Page 7). It seems that there are not enough data to support this conclusion. Transcriptome was detected at only one time point (Figure S3). RNA polymerase is a global transcription regulator and recognizes various promoters in genome. One concern is that the expression of genes is not linear with the concentration of RNA polymerase due to varied strength of promoters in genome. This may change cell physiology. So, more physiological tests (such as specific substrate uptake rate, maintenance coefficient, ATP yields et al) are expected. Also, an example using this growth-rate controller system will attract more audience.

(6) In the legend of Figure 4 (page 12), the "form" in "Six hours after removing IPTG form the medium..." should be "from"?

(7) Page 1, author list, "Song Xiaohu" should be "Xiaohu Song"

(8) Page 6, line 7-8, "Figure 6" should be "Figure 5"

Reviewer #3:

The authors describe and characterize an E coli strain carrying an inducible RNAP system. By adjusting the level of induction, growth rate is modulated from slow to nominal growth rates.

Major comment -

As the authors discuss, the cell composition correlates strongly with the growth rate; however the details of the correlation depend upon how the growth rate is changed. For example, under changes in growth rate modulated by nutrient change, the RNA content (indicative of ribosome content)

correlates positively with growth rate. For growth rate modulated by translation-targeting antibiotics, the RNA content correlates negatively. For growth rate modulated by transcription-targeting antibiotics (eg. rifampicin), the RNA content is virtually independent of growth rate (Scott et al (2010)).

In order to better understand how the cell perceives RNAP-limitation, it would be useful to include how the total RNA correlates with growth rate in the exponentially-growing cultures at different inducer concentrations. For context, that should be compared to analogous inhibition by rifampicin. It is also important to include at least one more nominal growth condition (such as glucose and 0.2% casamino acids, or Neidhardt's rich defined medium) to distinguish RNAP-specific response from passive physiological response to nutrient change.

Without this kind of physiological characterization, it is difficult to see how the genetically-modified strain is superior to rifampicin treatment of a wildtype strain. [Rifampicin inhibition at subinhibitory levels is likewise reversible.]

Minor technical comments -

1. The system exhibits strong cooperativity ($n=4$) in the growth rate attenuation (Fig 1D). That is potentially attributable to lacY still being present in the strain. A more gradual tuning of the growth rate may be achieved if lacY is deleted. Kuhlman et al. (2007) found that the cooperativity dropped to $n=2.5-3$ for the native promoter when lacY is deleted; down to potentially $n=1-1.5$ with the promoter that is illustrated in Fig. 1A.
2. The growth rate of both the reference W strain and modified R strain is very low; about 2/3 what is typical of batch growth in glucose minimal media [about 0.57/h or 0.82 db/h is typical]. The growth in flasks (Fig. S3) provides no growth rate estimates, so it is difficult to compare. My own experience is that with mineral oil layered on microplate wells, growth is anaerobic (irrespective of the addition of beads). I know of no published control experiment to the contrary; perhaps the authors could provide a comparison of microplate and flask growth in the supplement to ensure the growth conditions are aerobic and consistent with typical growth rates obtained in previous studies.
3. (cosmetic) Figure 5 seems to have rendered incorrectly, with overlapping labels.

Response to Reviewers of "Growth-rate control by reengineering the regulation of the transcription machinery" (MSB-13-5001)

Jérôme Izard^{1,2,+}, Cindy D.C. Gomez Balderas^{1,2,+}, Delphine Ropers¹, Stephan Lacour^{1,2}, Xiaohu Song³, Yifan Yang³, Ariel B. Lindner³, Johannes Geiselmann^{1,2,*}, Hidde de Jong^{2,*}

1. Laboratoire Interdisciplinaire de Physique (CNRS UMR 5588), Université Joseph Fourier
140 rue de la physique BP 87, 38402 Saint Martin d'Hères France.
 2. INRIA Grenoble - Rhône-Alpes
655 avenue de l'Europe, Montbonnot, 38334 Saint Ismier Cedex, France.
 3. Center for Research and Interdisciplinarity, INSERM U1001
Medicine Faculty, site Cochin Port-Royal, University Paris Descartes, Sorbonne Paris Cité
24 rue du Faubourg Saint Jacques, 75014 Paris, France.

⁺ Both authors contributed equally to this work.

^{*} Corresponding authors with equal contributions:
Johannes Geiselmann (Hans.Geiselmann@ujf-grenoble.fr),
Hidde de Jong (Hidde.de-Jong@inria.fr)

In this document we describe how our resubmitted manuscript addresses the issues raised by the reviewers of our previous manuscript "Growth-rate control by reengineering the regulation of the transcription machinery" (MSB-13-5001). We would like to thank the reviewers for their comments and constructive criticism.

In order to address all issues brought up by the reviewers, we have redone many of the experiments reported in the original manuscript over the past 18 months, performed a large number of additional control experiments, and developed new strains and tests for providing a concrete example of the applicability of growth-rate control in a biotechnological context. Some of the results reported in the original manuscript were removed, in response to the reviewer comments and because the resubmitted manuscript emphasizes different aspects of the work. This change of scope is also reflected in the title of the resubmitted manuscript, which has become "A synthetic growth switch and its biotechnological applications". We estimate that more than half of the text and the figures in the manuscript are new, making this a genuine resubmission.

The resubmitted manuscript includes the following major changes:

1. We have redone all kinetic experiments and performed additional control experiments in flasks in order to exclude potential artifacts of growth in microplates. We have adapted the experimental protocols and developed new procedures for computing the growth rate. This has allowed us to establish that the strain with the external control of *rpoBC* expression functions as a growth switch.
2. We have quantified the relative concentration of RNA polymerase *in vivo* by constructing a reporter strain in which the limiting β' subunit has been tagged with the fluorescent protein

mCherry. This construct has made it possible to show that the switch is due to the strong, ultrasensitive response of the growth rate to the concentration of β' .

3. The *lac* induction system has been modified, by deleting the *lacY* gene, in order to show that the growth switch does not depend on the specific induction system used.
4. We have characterized the physiology of the growth-arrested strain by measuring the mass ratio of total RNA and protein.
5. We provide a proof-of-principle of the biotechnological applicability of the growth switch by endowing both the wild-type strain and the growth-controlled strain with the capacity to produce glycerol. We show that the growth-arrested strain remains metabolically active and that its glycerol production yield is much higher than in the wild-type strain.
6. The growth-arrest strategy of this paper has been compared with a classical approach based on the use of antibiotics (rifampicin). This has allowed us to show that the growth switch developed in this manuscript is more stable than the rifampicin-arrested wild-type strain.

Below we respond to each of the reviewer comments in detail and we summarize the changes made to the manuscript. The reviewer comments are in *italic*, and our response in default font. In this document, we use the same abbreviations for strains as in the manuscript: W (wild-type strain), R (strain with *rpoBC* expression under control of IPTG), W-*rpoC*-mCherry (W strain with protein fusion of β' and mCherry), R-*rpoC*-mCherry (R strain with protein fusion of β' and mCherry), R- Δ *lacY* (R strain with *lacY* deletion).

Editor

The reviewers raise significant concerns regarding the interpretation of the main findings and the conclusiveness of the study. In particular, they are not convinced that the presented experimental data provide sufficient support for the quantitative dependence of growth rate on RNA polymerase levels and they point out that in absence of a concrete example demonstrating the applicability and efficacy of the strategy, its relevance for biotechnological applications remains unclear. Moreover, they think that further experimental evidence is required to convincingly exclude the induction of stress responses and that several additional control experiments should be performed throughout the study.

Considering these rather substantial concerns, we feel that we have no choice but to return this manuscript with the message that we cannot offer to publish it.

Nevertheless, as the reviewers did have positive words for the goals and the potential relevance of the study for biotechnology, we would like to indicate that we would be willing to reconsider a new submission based on this work. Any new submission would need to include new experimental data rigorously addressing the concerns raised by the reviewers.

Answer: We have addressed the concerns expressed above. In particular, we have done additional experiments to show that the observed variation of the growth rate with the induction strength of the *rpoBC operon* is indeed due to variations in the level of the RNA polymerase subunits encoded by this operon. To this end, we constructed a chromosomal fusion of the *rpoC* gene with the gene encoding the mCherry fluorescent reporter protein, to determine *in vivo* the accumulation of β' for different concentrations of the inducer (IPTG). An interesting conclusion from

these experiments is that the growth-rate response to changes in β' is highly ultrasensitive, making our strain function as a growth switch (see the reply to the reviewers for more details). Moreover, we have done a large number of additional control experiments requested by the reviewers (see the list in the introduction and below) and we provide a proof-of-principle of the interest of our growth switch for biotechnological applications by showing that growth limitation can improve the yield of glycerol production. This new material has led to major changes in the manuscript that are detailed in the responses to the individual reviewer comments below.

Reviewer 1

In this work the authors propose a way for externally controlling the cellular growth rate of E. coli. This is achieved by replacing the promoter for the $\beta\beta'$ RNA polymerase subunits (encoded by rpoB and rpoC) by an inducible promoter, based on the lac system, and responsive to the synthetic compound IPTG. This allows the authors to achieve growth-rates for E. coli spanning over two-fold in range, without changing the composition of the growth media. The authors suggest this method for growth-rate modulation irrespective of media composition may serve as a valuable tool for studying bacterial physiology and advocate its utility for biotechnological applications.

The ability to externally control the growth rate of microorganisms is indeed a highly desirable trait in both the study of cellular physiology and in biotechnology, and as such this work poses an ambitious and important goal. Also, achieving this goal by controlling the concentrations of RNA polymerase is an interesting avenue of research, as this may alter the growth rate without disrupting the metabolic capabilities of the cells, which is crucial for the utility of this approach for biotechnology. However, I was not convinced that the experimental procedures undertaken by the authors to address this challenge are adequate for several reasons, as detailed below, and therefore highly doubt that the constructed strain may actually serve as a tool for both basic research and biotechnology.

First, the authors chose to modulate $\beta\beta'$ levels by an IPTG-inducible promoter, similar to the well-studied endogenous lac promoter. Being a model system for transcriptional regulation, much work has been dedicated to characterizing the induction curve of the lac promoter in response to external IPTG. Much of this work showed that lac expression in the population behaves like a switch - cells are either fully ON or fully OFF. As such, intermediate levels of IPTG do not induce intermediate levels of induction in all cells, but rather change the fraction of cells that are ON (see Novick and Weiner 1957, Ozbudak et al. 2004, Vilar et al. 2003, Robert et al. 2010 and references within). Notably, Izard et al. also observe this phenomenon in their synthetic construct when treating the cells with intermediate levels of IPTG (30 μ M) and observing that the population divides into two subpopulations displaying vary different morphology and growth rate (Supplementary figure 5). As such, it appears to me that the main goal of the authors, which is to modulate growth rate by varying levels of RNA polymerase for individual cells, has not been achieved by this methodology. Personally, I would be very apprehensive about any conclusion about growth rate and physiology which is based on this system, which generates mixed populations of growing and non-growing cells.

Answer: In the original manuscript we claimed that the strain allowed a graded control of the growth rate over a large range, thus making it behave like a rheostat (Rossi *et al.*, 2000). The comments of this reviewer and the other reviewers have profoundly changed our view of the functioning of our strain. We have redone all kinetic experiments and performed additional control experiments, including quantification of the β' levels in the cell using a fluorescent reporter and

experiments with $\Delta lacY$ strains, and revised all data analysis procedures.

From the results of these experiments and analyses, we conclude that our strain does not behave like a rheostat but like a switch, in the sense that when increasing the IPTG concentration in the medium, above a certain threshold the growth rate in exponential phase rapidly switches from 0 to a value close to the maximum supported by the medium. We show that this switching behavior is not due to the cooperative response of the *lac* induction system, since the switching behavior persists in a $\Delta lacY$ strain (R- $\Delta lacY$). This strain eliminates most if not all of the cooperativity of the response of the promoter activity to changes in the concentration of the inducer, IPTG (see also the comments of Reviewer 3). The observed ultrasensitive response is therefore not due to the induction system used. Instead, we show that the switching phenotype arises from the response of the growth rate to the β' concentration, by quantifying the *in-vivo* concentration of the β' subunit using a fluorescently tagged RNA polymerase constructed for the purpose. We are working on elucidating the architecture of the network that leads to ultrasensitivity, but this topic is beyond the scope of the present manuscript.

The state of growth arrest of the R strain, when IPTG concentrations are too low to produce $\beta\beta'$ at a level sufficient for growth, proves to be quite stable, in the sense that even after 24 h in minimal M9 medium with glucose, no escape from growth arrest is observed for IPTG concentrations below 30 μM (Figure 2, see also Supplementary Figure S16). This shows that population heterogeneity involving growing and non-growing cells is negligible for the R strain at low IPTG concentrations, since even a small proportion of growing cells would have taken over the population after 25-30 generations. The reviewer is right that heterogeneity of the growth phenotype may occur at the switching threshold, between 20 and 30 μM IPTG for the R strain growing in minimal medium with glucose. However, given that the growth switch is expected to operate in either the ON or the OFF state, as shown for the biotechnological applications reported in the new manuscript, we do not consider this situation problematic.¹

In conclusion, in the resubmitted manuscript we argue that the R strain functions as a growth switch, rather than as a graded growth-rate controller. In addition to the fundamental biological applications that can be imagined for such a growth switch, we demonstrate its biotechnological applicability in a pilot study in which both the W and R strains have been equipped with plasmids for glycerol production. We show that the growth-arrested R strain continues to produce glycerol, at a yield that is two times higher than in the W strain and close to the theoretical maximum yield.

Action taken: We present the experiments supporting our modified view of the growth control exerted by our strain (switch rather than rheostat) in the section *Reengineering transcriptional control of RNA polymerase*, including the experiments with the $\Delta lacY$ strain (Figure 2). The stability of the R strain is discussed in detail in the section *Stability of the growth switch* of the *Discussion*, where we also report additional results about the long-term stability of the strain (Supplementary Figure S16). The quantification of RNA polymerase by means of the mCherry tag is reported in the section *Quantitative dependence of the growth rate on the concentration of RNA polymerase* and in Figure 3, while the biotechnological application of the growth switch is discussed in the section *Growth switch enables the improvement of metabolic production yields* and in Figure 5.

Second, while biotechnological applications are stated to be a strong motivation for this study,

¹In the original manuscript, we also reported some heterogeneity in cell size at 30 μM (Figure S5 in the original manuscript) and we associated this with bistability due to the *lac* induction system. In the light of the data presented in the new manuscript, notably the experiments with the R- $\Delta lacY$ strain, this conclusion was wrong. In addition, careful quantification of the cell size at this concentration (Figure S13 in the new manuscript) shows that the cell size distribution, although definitely broader at lower IPTG concentrations, remains unimodal.

there is no experiment showing the biotechnological efficacy of this technique, not even as a proof of concept. I think this proof of concept experiment is important as a priori the experimental system appears to have several inherent limitations, which may inhibit its use in biotechnological settings. For example, the authors suggest that this system will enable to maintain slow growth rates, even in nutrient-rich environments, which is important for sustaining high metabolic rates. This is a very unstable situation, as any cell that due to mutation is able to circumvent the repression and grow faster, will quickly overtake the population and the advantage of the system will be lost. The authors provide a partial response to this issue by inserting to the genome two additional copies of the repressor lacI. Whereas this indeed reduces the probability of escaping repression by mutating lacI, it does not deal with other escape options such as mutating the lacO sites, duplicating the entire genomic region etc. I was not convinced that this system may indeed prove stable in the long-term growth periods required for biotechnology.

Answer: In order to show the practical usefulness of the growth-rate controller for biotechnological applications, we followed the suggestion of the reviewer and developed a proof-of-principle. We constructed a plasmid enabling the W and R cells of *E. coli* to produce glycerol while growing on glucose, giving rise to strains that we called W-gly and R-gly, respectively. We show that growth-arrested R-gly cells remain metabolically active and are able to produce glycerol at a twice higher yield than in the W-gly strain grown in the same conditions. Remarkably, the glycerol production yield in the R-gly strain is close to the maximum theoretical yield computed from flux balance models of *E. coli* metabolism.

In addition, we have tested the genetic stability of our construction in experiments in which R cells were grown in minimal medium with glucose in a microplate under limiting conditions (0 μ M IPTG). We observed no escape after 24 h of growth on minimal medium with glucose and after 48 h only 6% of the cultures had resumed growth. For many biotechnological applications, the R strain thus meets basic stability requirements. Notice that, when necessary, strains with even stronger stability properties could be designed by developing alternative induction systems. We are currently pursuing this possibility, but this perspective is beyond the scope of the present manuscript.

Action taken: We added a section on the biotechnological proof-of-principle in the new manuscript (*Growth switch enables the improvement of metabolic production yields*) and summarize the results in Figure 5. We discuss the genetic stability of our construction in the *Discussion* section, where we also briefly summarize the results from the long-term growth experiments (the data are shown in Supplementary Figure S16).

Altogether, I think that the goal of experimentally modulating cell growth by changing RNA polymerase levels in cells is interesting. It would be very interesting to determine the quantitative dependence of growth rate on RNA polymerase levels. However, it appears that the genetic construct presented in this manuscript does not allow to properly address this issue. This construct does provide other interesting results, such as the morphology of the cells when RNA polymerase is repressed, yet these may be better presented in a different context asking different questions than those currently posed in the introduction.

Answer: In the resubmitted manuscript we have quantified the dependence of the growth rate on RNA polymerase levels using the mCherry tag to β' (see above and point 3 below). We have followed the suggestion of the reviewer to give less emphasis to the interesting observation that for low expression levels of *rpoBC* changes in cell morphology are observed. Instead, the biotechnological interest of our growth switch has been further developed.

Action taken: We have added a section on the precise quantification of the dependence of the growth rate on β' , in which we have also included the Western blots from the original manuscript and measurements of the total RNA and protein contents in the W and R strains (*Quantitative dependence of the growth rate on the concentration of RNA polymerase*).

Additional comments:

1. *The methods do not discuss how μ_{max} is calculated from the growth curves. Accordingly, it is not shown that the cells display exponential growth for all conditions. This is of importance because the authors pre-grow their strains in a media fully induced with IPTG. Therefore, if not calculated carefully, μ_{max} may actually represent the residual growth with the RNA polymerase left in the cells from the previous culture conditions.*

Answer: The reviewer is right. In our conditions, when the IPTG concentration is limiting, the growth rate initially decreases as the RNA polymerase from the preculture is diluting out. After this initial transient, however, the concentration of IPTG controls growth in a dose-dependent manner. Either the growth rate becomes 0, when the RNA polymerase concentration is too low to sustain growth, or the growth rate reaches the value that characterizes exponential growth in the given medium at the specific IPTG concentration. In the new manuscript, we redid all kinetic experiments and report the growth rate that is reached once the effects of the preculture have been lost (something we have not systematically done in the previous version of the manuscript). In this context, the use of the term μ_{max} is misleading and we have avoided it in the new manuscript.

Action taken: Given that the point raised by the reviewer is crucial, we explain in detail the procedure for performing the kinetic experiments and for computing the growth rate in the section *Reengineering transcriptional control of RNA polymerase...* and in the *Materials and methods*. All growth kinetics have been redone for this manuscript and all data have been treated following the procedure. The new results, displaying the switching phenotype, are reported in Figure 2 and in Supplementary Figure S6.

2. *The authors claim that their methods can span a large range of growth rates, from (almost) zero to the maximal growth rate possible in the tested environmental conditions. However, in figure 1C,D: it appears that with concentrations $< 30 \mu\text{M}$ the strain does not exhibit balanced exponential growth. From $30 \mu\text{M}$ to $1000 \mu\text{M}$ the cells span only two-fold in growth rate. In most experiments in which growth rate is modulated by media composition, a span of at least four-fold is easily achievable. A span of two-fold, even if it was generated by a homogenous population, is a rather small range if one wants to study bacterial physiology.*

Answer: This comment and the previous comment of the reviewer have changed our view on the behavior of our strain in important ways. The new growth kinetics data and the procedure for computing the growth rate show that our strain functions as a switch rather than providing a graded growth-rate response. As explained above, in our new manuscript we have shown that this switching behavior arises from the ultrasensitivity of the growth rate to changes in the β' concentration and demonstrated the biotechnological potential of the growth switch.

3. *The authors show how growth rate is dependent on IPTG concentrations (figure 2) and sev-*

eral points showing the dependence of β' abundance on IPTG. From a basic science perspective, it would be very interesting to characterize the dependence of growth rate on β' abundance. Is this relationship linear? (See Dykhuizen *et al.*, 1987 for a similar analysis done for the *lac* genes or Rest *et al.*, 2012 for this analysis on a yeast gene).

Answer: For the new manuscript, we have constructed a variant of the W and R strains carrying a chromosomal fusion of the *rpoC* gene with the gene encoding the mCherry fluorescent reporter protein (the W-*rpoC*-mCherry and R-*rpoC*-mCherry strains, respectively). The tagged β' subunit encoded by this construction allows the *in-vivo* quantification of RNA polymerase (Bratton *et al.*, 2011). The W-*rpoC*-mCherry and R-*rpoC*-mCherry strains have been used to quantify the dependence of the growth rate on the β' subunit of RNA polymerase. This has led to the interesting and unexpected observation that the growth-rate response to changes in β' is highly ultrasensitive. Moreover, our data suggest that RNA polymerase concentrations in wild-type *E. coli* have been optimized so as to ensure maximal growth with minimal investment in the synthesis of β and β' subunits. In the words of Rest *et al.* (2013), the expression of *rpoBC* in *E. coli* seems to be perched "on the edge of a fitness cliff".

Action taken: The results obtained with the W-*rpoC*-mCherry and R-*rpoC*-mCherry strains are reported in the section *Quantitative dependence of growth on the concentration of RNA polymerase* of the new manuscript. We put the results in the context of the work of Dykhuizen *et al.* and Rest *et al.* in the *Discussion* section.

4. *The analysis of gene expression seems rather superficial and not convincing. The authors should report the number of differentially-expressed genes and display the results also in a common graphical representation (e.g. heat map or scatter plot). Also, there are many issues with analyzing genome-wide expression datasets when global changes (such as changes in total RNA content) are at play, as presumably is the case in this experiment (Loven et al., 2012). There are ways to address these issues experimentally (Loven et al., 2012; Bakel and Holstege, 2008; van de Peppel et al, 2003; Islam et al, 2011), however it does not appear that any such controls were performed by the authors.*

Answer: We have become increasingly aware that more stringent controls are needed to draw stronger conclusions from the RNA-Seq data presented in the previous manuscript. We have therefore decided not to include the RNA-Seq data in the resubmitted manuscript and to perform new experiments with appropriate controls for a follow-up paper. The removal of the RNA-Seq data is more than compensated by new material in the other sections and especially the proof-of-principle of the use of the growth switch for biotechnological applications.

Action taken: The section on the RNA-Seq experiments has been removed from the resubmitted manuscript.

5. *In the introduction there are some strong statements not supported by data. The early Copenhagen works showed that the physiological properties of the cell go together with growth rate. The authors state that these properties are determined by growth rate, which implies a causality that is highly debatable in the field (see Levy and Barkai, 2009 for evidence that the causality may be reversed). Accordingly, the authors declare that these properties are optimized, again a strong claim which is not claimed in the original papers and demands proof.*

Answer: Some statements in the introductory paragraph, summarizing the background of our

work, were indeed too approximate or cavalier. We have reformulated the paragraph, indicating that growth rate and physiological properties are correlated (the direction of the causal arrow is indeed subject of debate). We have removed the phrase that these physiological properties have been optimized so as to achieve the maximum possible growth rate within a given medium, which is indeed too general when formulated in this way and not claimed as such in the original papers.

Action taken: The introductory paragraph has been rewritten.

Reviewer 2

*The authors described the reversible control of cell growth rate by replacing the *E. coli* native RNA polymerase *rpoBC* promoter with an IPTG-inducible promoter. By tuning the concentration of the inducer IPTG, the authors successfully tuned the *E. coli* growth rate between zero and the maximal achievable growth rate. The authors further demonstrated that this growth-arrest state could be reversed by external supplementation of IPTG. In the absence of IPTG, the growth-arrest cells continue maintaining metabolic activity but undergoing no cell division – an observation that is related with abnormal cell morphology and elongated cell shape. While these findings potentially could be important to optimal growth-rate control in synthetic biology and biotechnological applications, the scientific facts and the experimental data present in this study only weakly or indirectly support their major conclusion.*

Specific comments:

(1) The authors described that the intracellular concentration of the β and β' subunits relative to the α subunit were determined (page 5). However, only the result of β' subunit was displayed (Fig. 2). The genes encoding β and β' subunits were located in an operon. Generally speaking, the gene proximal to the promoter in an operon transcribes more RNA than those behind it. But, translation may change protein contents in cells. So, the authors should give an explanation why the amount of β' subunit sets the overall concentration of RNA polymerase but not β subunits. Otherwise, the relative concentration of β subunits should be determined.

Answer: The reviewer is right that we only quantified the concentration of the β' subunit (relative to the α subunit in the previous manuscript and also directly using a fluorescent reporter in the new manuscript). The reason for limiting the immunological quantifications to the β' subunit was purely technical. Despite many different attempts, we could not get the antibody directed against the β subunit to work properly. The implicit assumption we made is that the cellular concentrations of the β and β' subunits are (approximately) the same, given that the *rpoB* and *rpoC* genes are included in the same operon and no post-transcriptional regulatory mechanisms specific for the one or the other subunit are known. Interestingly, recent work using the ribosome profiling technology has shown that the synthesis rates of the β and β' subunits (in numbers of molecules per generation in balanced growth) are the same within the limits of experimental error ($< 10\%$) Li *et al* (2014). This validates our assumption.

Action taken: We have made explicit the assumption for measuring the β' subunit only and justified it by citing the result from the study of the Weissman lab (Li *et al*, 2014). Moreover, we have been more careful in choosing our wording and we no longer say that we quantified both

subunits, but that we quantified the β' subunit as representative of both the β and β' subunits.

(2) *As expected, the relative expression of the β' protein is decreasing when the authors reduced the amount of IPTG in the culture (Figure 2). However, the expression of the α subunit is also subject to the control of the functional RNA polymerase. The expression of the reference protein (α subunit) appears remained constant when the level of externally added IPTG changes. How could this happen?*

Answer: Notice that we loaded the lanes of the gel with equal amounts of protein, as stated in the caption of the figure. The fact that the quantity of α subunit does not vary across the conditions should therefore not be interpreted as constant expression of α . Rather, it means that α takes the same proportion of the total amount of protein across conditions and is thus expressed at the same relative rate as the total amount of bulk protein (although its absolute rate will vary across conditions). Similarly, the decrease of the quantity of β' protein means that, with lower IPTG concentrations, the proportion of β' within the total amount of protein becomes increasingly smaller. In our view, the ratio of the β' and α units is really the interesting quantity here, as it indicates that at lower IPTG concentrations β' becomes increasingly limited with respect to α .

Action taken: We better explain in the section *Quantitative dependence of the growth rate on the concentration of RNA polymerase* how the results in Figure 3 should be interpreted.

(3) *It is unclear how the growth rate of the wild type strain would change at different concentration of IPTG. The authors demonstrated that the specific growth rate remained constant for the wild type strain in the absence (0 mM) or the presence (1 mM) of IPTG in Figure 1D. Chemical inducers, however, are generally known as stress elicitors that, in some cases, could trigger global transcription changes that are detrimental to cell growth. The authors are suggested to provide more data to support their conclusion.*

Answer: We tested that the growth rate of the wild-type (W) strain does not change when adding no IPTG or when adding IPTG at the maximum concentration used in this study (1 mM). In our view, this excludes "global transcription changes that are detrimental to cell growth" in the range of IPTG concentrations considered. In the growth kinetics carried out for the new manuscript, we repeated these controls and also added IPTG at intermediate concentrations to provide further evidence, as suggested by the reviewer. In addition, we checked that varying the concentration of IPTG in the medium does not affect transcription from the natural *rpoBC* promoter in the wild-type strain, using the W-*rpoC*-mCherry strain. This demonstrates that the addition of IPTG does not affect transcription from the natural *rpoBC* promoter in our conditions.

Action taken: We added the data for the control experiments that test the effect of IPTG on the growth rate and transcription from the natural *rpoBC* promoter in the wild-type strain in Supplementary Figure S6. We discuss the results in the *Engineered control of RNA polymerase...* of the new manuscript.

(3) *The growth kinetics obtained in microplates (Figure 1C) and microfluidic channels (Figure 4) are not enough to draw all the conclusions. Due to edge effects and transport limitations, cell growth rate estimated with these devices are generally considered as not accurate as compared with flask or chemostat experiments. The growth curves were somewhat different in microplate and flask for R strain and W strain under 1 mM IPTG (Figure 1C and Figure S3). It is safe to test the*

growth of R strain under all IPTG concentrations (as shown in Figure 1C) in flask instead of only 0 μ M, 30 μ M and 1 mM.

Answer: In order to test that the data from microplate and flask experiments are comparable, we have systematically and extensively compared the microplate experiments with corresponding flask experiments and observed that the growth rates in the two conditions agree very well. Following the comments of this reviewer and Reviewer 3, we have also redone all microplate growth experiments in conditions that improve aeration (adding glass beads for stirring, avoiding use of mineral oil, high frequency of shaking). We have used 5 (biological) replicates on average to compute growth rates and we have eliminated data from wells showing edge effects. Moreover, when computing growth rates in exponential phase, we only use data below an absorbance of 0.2 (corresponding to an optical density of 0.45), so as to avoid growth-limiting oxygen transfer rates.

Action taken: The data from the growth kinetics in microplates and flasks are reported in Figure 2 and discussed in the section *Reengineering transcriptional control of RNA polymerase* of the main text. The experimental conditions are motivated in the same section and described in detail in the *Materials and methods*.

(4) *Cell growth rate are generally related with the amount of ribosomal proteins and the availability of cellular resources (such as amino acids) inside the cell. In order to refine this article, the authors are suggested to quantify the amount of ribosomal proteins and ribosomal RNAs (rRNAs) at different level of IPTG.*

Answer: As an additional indicator of how the growth switch affects global cell physiology, we have quantified the total amounts of RNA and protein in growth-arrested and growing R and W strains (see also the comments of Reviewer 3). The mass ratio of total RNA and protein is often used as a proxy for the protein synthesis rate and for the active ribosome concentration (Scott *et al*, 2010). We notably compared these measurements with those reported by Scott *et al* (2010) in their Figure S2, where they observed that inhibition of transcription by rifampicin decreased the growth rate, but did not change the mass ratio of total RNA and protein, contrary to what is observed for translation inhibition by chloramphenicol, where this ratio increases. Interestingly, when comparing growth-arrested and growing R and W strains, we measured a constant mass ratio of total RNA and protein. This observation provides additional evidence that growth arrest in the absence of IPTG is indeed due to the repression of the *rpoBC* genes and the resulting lack of RNA polymerase available for transcription.

Action taken: The results of the protein and RNA quantification experiments are reported in the section *Quantitative dependence of growth on the concentration of RNA polymerase* and discussed in the context of the work of Scott *et al* (2010).

(5) *"It is reassuring that the chosen method for growth limitation does not trigger a major stress response or other profound physiological modifications" (Page 7). It seems that there are no enough data to support this conclusion. Transcriptome was detected at only one time point (Figure S3). RNA polymerase is a global transcription regulator and recognizes various promoters in genome. One concern is that the expression of genes is not linear with the concentration of RNA polymerase due to varied strength of promoters in genome. This may change cell physiology. So, more physiological tests (such as specific substrate uptake rate, maintenance coefficient, ATP yields et al) are expected. Also, an example using this growth-rate controller system will attract more audience.*

Answer: We have become increasingly aware that more stringent controls are needed to draw stronger conclusions from the RNA-Seq data presented in the previous manuscript. We have therefore decided not to include the RNA-Seq data in the resubmitted manuscript and to do new experiments with appropriate controls for a follow-up paper. On the other hand, following the suggestion of this reviewer and the other reviewers, we have developed a proof-of-principle of the practical interest of the growth switch for biotechnological applications. We transformed W and R cells of *E. coli* with a plasmid that allows the production of glycerol when the cells grow on minimal medium supplemented with glucose, giving rise to the W-gly and R-gly strains, respectively. We show that growth-arrested R-gly cells remain metabolically active and are able to produce glycerol at a twice higher yield than the W-gly strain grown in the same conditions. Remarkably, the glycerol production yield in the R-gly strain is close to the maximum theoretical yield that can be attained by *E. coli* cells.

Action taken: The section on the RNA-Seq experiments has been removed from the resubmitted manuscript. We added a section on the biotechnological proof-of-principle in the new manuscript (*Growth switch enables the improvement of metabolic production yields*) and summarize the results in Figure 5.

(6) *In the legend of Figure 4 (page 12), the "form" in "Six hours after removing IPTG from the medium..." should be "from"?*

Action taken: Corrected.

(7) *Page 1, author list, "Song Xiaohu" should be "Xiaohu Song"*

Action taken: Corrected.

(8) *Page 6, line 7-8, "Figure 6" should be "Figure 5"*

Action taken: Figure 5 is no longer included in the new manuscript.

Reviewer 3

The authors describe and characterize an E coli strain carrying an inducible RNAP system. By adjusting the level of induction, growth rate is modulated from slow to nominal growth rates.

Major comment:

As the authors discuss, the cell composition correlates strongly with the growth rate; however the details of the correlation depend upon how the growth rate is changed. For example, under changes in growth rate modulated by nutrient change, the RNA content (indicative of ribosome content) correlates positively with growth rate. For growth rate modulated by translation-targeting antibiotics, the RNA content correlates negatively. For growth rate modulated by transcription-targeting antibiotics (eg. rifampicin), the RNA content is virtually independent of growth rate (Scott et al (2010)).

In order to better understand how the cell perceives RNAP-limitation, it would be useful to include how the total RNA correlates with growth rate in the exponentially-growing cultures at different inducer concentrations. For context, that should be compared to analogous inhibition by rifampicin. It is also important to include at least one more nominal growth condition (such as glucose and 0.2% casamino acids, or Neidhardt's rich defined medium) to distinguish RNAP-specific response from passive physiological response to nutrient change.

Answer: This point is similar to the one raised by reviewer 2. We repeat the answer here. Following the suggestions of the two reviewers we quantified the total amount of RNA and protein in growth-arrested and growing R and W strains. We compared these measurements with those reported by Scott *et al* (2010) in their Figure S2, where they observed that inhibition of transcription by rifampicin decreased the growth rate, but did not change the mass ratio of total RNA and protein, contrary to what is observed for translation inhibition by chloramphenicol, where this ratio increases. Interestingly, when comparing growth-arrested and growing R and W strains, we measured a constant mass ratio of total RNA and protein. This observation provides additional evidence that growth arrest in the absence of IPTG is indeed due to the repression of the *rpoBC* genes and the resulting lack of RNA polymerase available for transcription.

In addition to minimal M9 medium with 0.2% glucose, we also tested the growth switch in other growth media (minimal M9 medium with 0.2% glucose and 0.2% casaminoacids, minimal M9 medium with 0.2% succinate, and rich LB medium with 0.2% glucose). The growth switch functions in all of these media. Interestingly, as the medium becomes richer, supporting a higher maximum growth rate, the IPTG threshold at which the switch occurs increases as well.

Action taken: The quantification of total RNA and protein is reported in the section *Quantitative dependence of growth on the concentration of RNA polymerase*, where the results are compared with those of Scott *et al* (2010), as suggested by the reviewer. The functioning of the growth switch in other media is reported and discussed in the section *Reengineering transcriptional control of RNA polymerase and in the Discussion..*

Without this kind of physiological characterization, it is difficult to see how the genetically-modified strain is superior to rifampicin treatment of a wildtype strain. [Rifampicin inhibition at subinhibitory levels is likewise reversible.]

Answer: The original motivation for using transcriptional control of the *rpoBC* operon is that, in principle, the *lac* system used in this study can be replaced by other induction systems. The latter might be coupled, for example, to intracellular inducer molecules sensing the cell physiology, in order to create more complex control structures. This synthetic biology perspective, which is quite different from rifampicin treatment, may not have been sufficiently emphasized in the original manuscript.

In addition, to compare the performance of our approach with classical approaches, we performed long-term growth experiments with both the growth-arrested R strain and the W strain treated with rifampicin. We found our growth switch to be more stable than the antibiotics-treated wild-type cells, in the sense that after 48 h of suspension in minimal medium with glucose in a microplate under limiting conditions (0 μ M IPTG), only 2 out of the 33 cultures of the R strain had escaped, whereas almost half of the W cultures grown in minimal medium with rifampicin had escaped by that time.

Action taken: We compare our growth switch with growth arrest by means of antibiotics in the

section *Stability of the growth switch* of the *Discussion*. The results of the long-term growth experiments are presented in Supplementary Figure S16. The perspectives of integrating our growth-rate controller as a module of larger synthetic regulatory networks is developed in more detail in the section *Growth switch is modular: towards interfacing the growth switch with physiological signals* of the *Discussion*.

Minor technical comments:

1. *The system exhibits strong cooperativity ($n=4$) in the growth rate attenuation (Fig 1D). That is potentially attributable to *lacY* still being present in the strain. A more gradual tuning of the growth rate may be achieved if *lacY* is deleted. Kuhlman et al. (2007) found that the cooperativity dropped to $n=2.5-3$ for the native promoter when *lacY* is deleted; down to potentially $n=1-1.5$ with the promoter that is illustrated in Fig. 1A.*

Answer: We remark that our view on the behavior of our strain has changed in important ways since the submission of the original manuscript, to a large extent thanks to the reviewer comments. As explained in the response to Reviewer 1, the new growth kinetics data and the procedure for computing the growth rate show that our strain functions as a switch rather than providing a graded growth-rate response over a large range. Moreover, using the mCherry tag of the β' subunit, we show that the observed cooperativity is due to the ultrasensitive response of growth rate to the concentration of RNA polymerase. As a consequence, within this new perspective, we do not expect an effect on the switching behavior of our strain when a *lacY* mutant is used. In order to test this, we followed the suggestion of the reviewer and constructed a variant of our induction system in which the *lacY* gene has been deleted (R- Δ *lacY* strain). When testing the response of the strain with this modified induction system, we indeed found that the system still functions as a growth switch, showing the independence of this phenotype from the specific induction system used.

Action taken: We present and discuss the results obtained with the R- Δ *lacY* strain in the section *Reengineering transcriptional control of RNA polymerase...* and in Figure 2 of the new manuscript.

2. *The growth rate of both the reference W strain and modified R strain is very low; about 2/3 what is typical of batch growth in glucose minimal media [about 0.57/h or 0.82 dbl/h is typical]. The growth in flasks (Fig. S3) provides no growth rate estimates, so it is difficult to compare. My own experience is that with mineral oil layered on microplate wells, growth is anaerobic (irrespective of the addition of beads). I know of no published control experiment to the contrary; perhaps the authors could provide a comparison of microplate and flask growth in the supplement to ensure the growth conditions are aerobic and consistent with typical growth rates obtained in previous studies.*

Answer: Following the comments of this reviewer and Reviewer 2, we have redone all microplate growth experiments, in conditions that improve aeration (adding glass beads for stirring, avoiding use of mineral oil, high frequency of shaking). We used 5 (biological) replicates on average to compute growth rates and we have eliminated data from wells showing edge effects. Moreover, when computing growth rates in exponential phase, we only use data below an absorbance of 0.2 (corresponding to an optical density of 0.45), so as to avoid growth-limiting oxygen transfer rates. In order to test that the data from microplate and flask experiments are comparable, as suggested by the reviewer, we have systematically and extensively compared the microplate experiments with corresponding flask experiments and observed very good correspondence of the growth rates within

our conditions.

Action taken: The data from the growth kinetics in microplates and flasks are reported in Figure 2 and discussed in the section *Reengineering transcriptional control of RNA polymerase....* The experimental conditions and procedures for computing the growth rate are motivated in the same section and described in detail in the *Materials and methods*.

3. (cosmetic) Figure 5 seems to have rendered incorrectly, with overlapping labels.

Answer: Figure 5 has been removed from the new manuscript (following the comments of Reviewers 1 and 2).

References

- Bratton BP, Mooney RA, Weisshaar JC (2011) Spatial distribution and diffusive motion of RNA polymerase in live *Escherichia coli*. *J Bacteriol* **193**: 5138–46
- Li GW, Burkhardt D, Gross C, Weissman JS (2014) Quantifying absolute protein synthesis rates reveals principles underlying allocation of cellular resources. *Cell* **157**: 624–35
- Rest JS, Morales CM, Waldron JB, Opulente DA, Fisher J, Moon S, Bullaughey K, Carey LB, Dedousis D (2013) Nonlinear fitness consequences of variation in expression level of a eukaryotic gene. *Mol Biol Evol* **30**: 448–56
- Rossi FMV, Kringstein AM, Spicher A, Guicherit OM, Blau HM (2000) Transcriptional control: rheostat converted to on/off switch. *Mol Cell* **6**: 723–8
- Scott M, Gunderson CW, Mateescu EM, Zhang Z, Hwa T (2010) Interdependence of cell growth and gene expression: origins and consequences. *Science* **330**: 1099–1102

Thank you again for submitting your work to Molecular Systems Biology. In addition to the previous reviewers #1 and #3 who were asked to examine whether their previous comments have been addressed, we invited a new referee (#2), who evaluated the study afresh. We have now heard back from these three referees. As you will see from the reports below, the referees raise a series of (mostly minor) concerns, which should be carefully addressed in a revision of the manuscript.

On a more editorial level, we would like to suggest a modified title i.e. "A synthetic growth switch based on controlled expression of RNA polymerase", in order to describe in better detail the key aspects of the work. Moreover, we think that the Discussion section is rather long and needs to be streamlined.

REFEREE REPORTS

Reviewer #1:

The authors present an engineered *E. coli* strain in which an inducible promoter was placed upstream of RNA polymerase. They characterize both the growth rate and RNA polymerase expression in response to external concentrations of IPTG and find that at high polymerase levels (high IPTG) the strain grows similarly to the wt, while at low levels it arrests growth in a reversible manner. They advocate the use of growth arrested strains for biotechnology and provide a proof of concept for the utility of the approach by engineering *E. coli* to produce glycerol.

The authors have performed extensive revisions to the manuscript addressed the concerns I previously raised regarding the work by substantial modifications. They have added much-needed controls to existing experiments, several important new experiments and have changed to a large extent their interpretation of the data and the system. I believe this manuscript has greatly improved in level and will be of interest to the wide readership of MSB, after addressing a few minor comments:

1. In the section "Reengineering transcriptional control of RNA polymerase allows a medium independent growth switch", the authors state that the threshold is just above 20uM. They actually have no way to know this as they measure 20uM and 30uM. They should use the more conservative wording that the switching threshold is between 20 and 30uM.
2. In the same section the authors state that they redid the experiments with higher dilution of the initial pre-culture. This is a very good control for their claims. However, I could not find a description of this procedure in the methods, nor a figure showing this data.
3. Figure 2B- Grey line separating between 100uM and 1000uM is not straight. There are conventions for how to plot a gap in the axis (with a dashed line).
4. I am not sure that 'kinetic experiments' is the term that best describes these experiments. Consider changing to 'growth rate measurements' or the like.
5. The results regarding the stability of the system (figure S16) are very important, but are buried in the discussion. These should be properly presented in the results. The authors should be cautious regarding their statements. A growth experiment of 24 or 48 hours can hardly be regarded as 'long-term'. The authors should clearly state that they observe escape from the growth-arrested state and discuss future avenues for addressing this.

Reviewer #2:

This manuscript describes the control of *E. coli* growth by using inducible expression of the core RNA polymerase enzymes encoded by *rpoB* and *rpoC*. The authors showed a switch-like response of cell growth as the inducer (IPTG) concentration in the medium varied. They then showed that the

non-growing cells could be revived by cycling the IPTG conc and that this switch could be operated repeatedly. Finally the system was put into a biotech application by showing that the strain with growth switch produced converted glucose to glycerol more effectively than WT cells.

Overall this is a solid piece of experimental work. Even though it is not a surprise that repressing RNAP synthesis would lead to growth arrest, doing this well is not an easy task due to the ease of mutation. The authors' use of multiple *lacI* genes to combat this problem is clever and effective, at least for cultures grown in 96-well microplate. However, I have some questions and concerns regarding the solidity of the specific conclusions drawn from this study.

1. The authors claim a switch-like growth transition upon RNAP titration. While the switch-like response to IPTG titration itself is clear, the authors claim that this results from a switch-like response of growth to RNAP abundance (top of p.8). This claim is based on the plot of growth rates vs RpoC-mCherry level (Fig. 3B, 3D): the data is fit by Hill functions with Hill coeff of 10. I am not at all convinced of this claim: the important region is for mCherry "concentration" from 0 to 1000 (by the scale shown in Fig. 3B, 3D). This is also the range of the RNAP concentration present in WT cells (shown by the red points.) Over this range, the experiments have only 2-3 data points, one at high growth, one near zero growth, and one in between. It is not possible to make a conclusion on the abruptness of the switch based on these 2-3 points. [Visually, one is easily biased by the data with conc > 1000 units; but those are irrelevant as the authors themselves stated.] Adding to this problem, I do not believe the authors growth protocol can be expected to measure slow growth accurately - to see balanced growth with a doubling time of 3-4 hours, a much longer period (say 4-5x longer than the doubling time) is needed.

In short, the author's strain and growth protocol is not designed to reveal the quantitative relation between RNAP concentration and growth. A careful characterization of this relation will likely require a very different titratable expression system and focused on the endogenous range of RNAP concentrations. This is clearly not the focus area of this study devoted to synthetic applications. But the authors should not make a claim that is outside of the means of their system.

2. The authors explained the importance of controlling cell growth as a strategy to direct nutrient flux into the production of desired products (rather than cell mass) in biotech applications, and stated problems with previous attempts at this strategy. Based on the higher conversion efficiency of glucose to glycerol in the R-strain compared to the W-strain, the authors concluded that their inducible RNAP strain presents an effective strategy. I am not convinced of this claim for several reasons given below.

Fig. 5A, 5B are the key plots on which the conclusions on glucose-to-glycerol conversion efficiencies are based. I see multiple deficiencies with the comparison. First the grey-shaded regions used for evaluation are not chosen on equal footing: The shaded region for strain W corresponds to exponential growth while that for strain R corresponds to transition to non-growing phase. To make comparison on equal footing, the minimum the authors should do is to arrange their culture so that the two strains have similar saturating OD, and select regions from the same OD range, e.g., from 50% saturation to 90% saturation. [In order to monitor glycerol synthesis directly for the entire range, it will be useful to remove glycerol uptake from strain W.] The comparison can be made on a more equal footing if the saturation of OD can be arranged to result from something other than glucose running out. They can, e.g., supply antibiotics to the W-strain while supplying IPTG to the R-strain, or they can apply their titratable system to control an essential metabolic gene. It is only then statement on the advantage of various growth control strategies be made.

Indeed, controlling metabolism or switching metabolic genes have been used in the metabolic engineering community in the past as the authors described in the text. However, just because the R-strain can do well converting glucose to glycerol does not mean the other strategies could not do an equally good job at this particular task. As the authors mentioned, a big problem with the strategy of controlling growth via metabolism is the cell's ability to adjust to metabolic perturbations. A specific problem is the inhibition of carbon uptake when growth is arrested by, e.g., depriving cells of supply of auxiliary nutrients such as nitrogen or sulfur. An example is the inhibition of PTS sugar uptake systems by a build-up of pyruvate. In the glucose-to-glycerol study done in this work, incoming carbon is diverted to synthesizing glycerol from DHAP, thus possibly relieving the build up of pyruvate. It is thus conceivable that other modes of metabolic perturbations can do the job equally

well as controlling RNAP synthesis. The authors made no comments on why they think controlling RNAP synthesis would alleviate the cells from experiencing the same problems found by controlling metabolism.

Finally, direct inspection of Fig. 5B shows that the 2/3 of glycerol was actually produced during the slow down of cell growth, as is known for growth control by metabolic means. Glycerol synthesis essentially stopped 300 minutes beyond the grey zone. This is at variance with the authors goal (stated in introduction) that enzymes will continue to function after switching off growth, enabling high-yield production of metabolites of interest. Perhaps this 300-minute after growth ceases is longer than what is achievable by other strategies. If so, then this should be emphasized, backed up by direct comparison in the post-growth regime.

3. It is very nice to see the demonstration that cell growth can be revived after up to 8-hr of no growth when medium contained no IPTG. Microfluidic devices are appropriate for switching out of IPTG. The observation of cell elongation without division is also very interesting. For biotech applications however, it will be useful to know the stability of these cells, especially during the stressful period with no IPTG, in shake flasks where there are 1000x more cells than in 96-well microplates (and 10^6 more than then 950 tracks followed in microfluidic devices), and hence with much higher rate of mutation to overcome the repression of RNAP synthesis.

Reviewer #3:

The authors have addressed any concerns I had with the previous submission. I have only very minor comments.

Minor points:

- "These experiments have shown that cell composition is correlated with the growth rate and not the nature of the specific nutrients available for growth"

This is only approximately true; see Ehrenberg, Bremer and Dennis (2013) "Medium-dependent control of the bacterial growth rate" *Biochimie* 95:643-58. PMID:23228516

-p3. regain -> regained

-p6. "this phenotype is therefore independent of the specific induction system used."

The control is a lacY deleted mutant still employing the lac-based expression system. The claim that the phenotype is independent of the specific induction system is strong; perhaps what was meant was "this phenotype is therefore not attributable to positive feedback in the inducer (IPTG) uptake."

-p7. "the sensitivity of the growth rate to the IPTG concentration is much lower at higher growth rates." Not clear what is meant by sensitivity here. Is this the slope of the interpolating curve on a log-log plot? Fit with a Hill function, the sensitivity (i.e. Hill coefficient) appears, by eye, to be roughly the same for all curves.

-Fig 4. A scale bar in the image panels would be helpful.

-p9. "growing to a length of about 20 μm 6 hours after IPTG removal."

This extreme filamentation suggests induction of the SOS response, or a transcriptional toxin/antitoxin pair. Although not directly relevant, it should perhaps be remarked in the text that the growth arrest appears filamentous.

Response to Reviewers of "A synthetic growth switch based on controlled expression of RNA polymerase" (MSB-15-6382)

Jérôme IZARD^{1,2,+}, Cindy D.C. Gomez Balderas^{1,2,+}, Delphine Ropers¹, Stephan Lacour^{1,2}, Xiaohu Song³, Yifan Yang³, Ariel B. Lindner³, Johannes Geiselmann^{1,2,*}, Hidde de Jong^{2,*}

1. Université Grenoble Alpes, Laboratoire Interdisciplinaire de Physique (CNRS UMR 5588)
140 rue de la physique BP 87, 38402 Saint Martin d'Hères France.
2. INRIA, Research center Grenoble - Rhône-Alpes, 655 avenue de l'Europe, Montbonnot
38334 Saint Ismier Cedex, France.
3. Center for Research and Interdisciplinarity, INSERM U1001
Medicine Faculty, site Cochin Port-Royal, University Paris Descartes, Sorbonne Paris Cité
24 rue du Faubourg Saint Jacques, 75014 Paris, France.

⁺ Both authors contributed equally to this work.

^{*} Corresponding authors with equal contributions:
Johannes Geiselmann (Hans.Geiselmann@ujf-grenoble.fr),
Hidde de Jong (Hidde.de-Jong@inria.fr)

In this document we describe how our revised manuscript addresses the issues raised by the reviewers of "A synthetic growth switch and its biotechnological applications" (MSB-15-6382). We would like to thank the reviewers for their comments and constructive criticism.

In order to address all issues brought up by the reviewers, we have added further data, performed an additional control experiment, and revised the text of the manuscript. The major changes are the following:

1. We confirmed the stability of the growth switch in a control experiment using flasks instead of microplates;
2. We have added a new subsection on the genetic stability of the strain, moving material from the *Discussion* and the Supplementary Information (now called Appendix) to the *Results* section. This new subsection also reports the results of the above-mentioned flask experiment;
3. We have shortened and streamlined the rest of the *Discussion*;
4. We report data on a control experiment in which bacteria were inoculated at a dilution ratio of 0.001 rather than 0.01;
5. We have reformulated and clarified some of the conclusions of the biotechnological application to avoid misinterpretation of our claims;
6. We have moved some figures from the Supplementary Information (now called Appendix) to the Expanded View, and renumbered the Table and the Movie to Appendix Table S1 and

Appendix Movie S1, respectively. Notice that we have maintained the Appendix Texts in the ordering of Appendix Figures, in order to avoid inconsistent numbering of figures in the Appendix Texts.

Below we respond to each of the reviewer comments in detail and we summarize the changes made to the manuscript (major changes in red). The reviewer comments are in italic, and our response in default font. In this document, we use the same abbreviations for strains as in the manuscript: W (wild-type strain), R (strain with *rpoBC* expression under control of IPTG), W-*rpoC*-mCherry (W strain with protein fusion of β' and mCherry), R-*rpoC*-mCherry (R strain with protein fusion of β' and mCherry), R- $\Delta lacY$ (R strain with *lacY* deletion).

Editor

Thank you again for submitting your work to Molecular Systems Biology. In addition to the previous reviewers 1 and 3 who were asked to examine whether their previous comments have been addressed, we invited a new referee (2), who evaluated the study afresh. We have now heard back from these three referees. As you will see from the reports below, the referees raise a series of (mostly minor) concerns, which should be carefully addressed in a revision of the manuscript.

On a more editorial level, we would like to suggest a modified title i.e. "A synthetic growth switch based on controlled expression of RNA polymerase", in order to describe in better detail the key aspects of the work. Moreover, we think that the Discussion section is rather long and needs to be streamlined.

Answer: We have changed the title as suggested, and shortened and streamlined the *Discussion* section. In particular, the subsection *Stability of the growth switch* has been moved to the *Results* section, following the suggestion of Reviewer 1, and has been enriched by data from the control experiment proposed by Reviewer 2. The new *Results* subsection is called *Stability of the growth switch* (p 11-12).¹ In addition, in response to comments from Reviewer 2, we have eliminated the subsection *Ultrasensitive response of growth rate to changes in RNA polymerase concentration* from the *Discussion*, moving some of the material to the subsection *Medium-independent and reversible growth switch* of the *Discussion* (p 13, par 1-2). Elsewhere in the *Discussion*, several phrases have been shortened to avoid unnecessary repetition of the main text (p 12, par 4; p 13, par 2-3).

Reviewer 1

The authors present an engineered E.coli strain in which an inducible promoter was placed upstream of RNA polymerase. They characterize both the growth rate and RNA polymerase expression in response to external concentrations of IPTG and find that at high polymerase levels (high IPTG) the strain grows similarly to the wt, while at low levels it arrests growth in a reversible manner. They advocate the use of growth arrested strains for biotechnology and provide a proof of concept for the utility of the approach by engineering E.coli to produce glycerol.

¹The changes in the main text are indicated by page number (p) and paragraph number (par). The first paragraph on a page may have begun on the previous page.

The authors have performed extensive revisions to the manuscript addressed the concerns I previously raised regarding the work by substantial modifications. They have added much-needed controls to existing experiments, several important new experiments and have changed to a large extent their interpretation of the data and the system. I believe this manuscript has greatly improved in level and will be of interest to the wide readership of MSB, after addressing a few minor comments:

1. In the section "Reengineering transcriptional control of RNA polymerase allows a medium independent growth switch", the authors state that the threshold is just above 20uM. They actually have no way to know this as they measure 20uM and 30uM. They should use the more conservative wording that the switching threshold is between 20 and 30uM.

Action taken: Done at several places in the text (p 6, par 2; p 7, par 1 and 5; p 20, par 2).

2. In the same section the authors state that they redid the experiments with higher dilution of the initial pre-culture. This is a very good control for their claims. However, I could not find a description of this procedure in the methods, nor a figure showing this data.

Action taken: We have completed a phrase in the *Materials and methods* describing the procedure (which is the same as for the growth experiments with the reference dilution factor) (p 16, par 2). The results are shown in the new panel *E* of Figure EV1.

3. Figure 2B- Grey line separating between 100uM and 1000uM is not straight. There are conventions for how to plot a gap in the axis (with a dashed line).

Answer: In fact, at first we had plotted the results as suggested by the reviewer, but we found the separation not clear enough (especially after having added the dashed curve and in the case of LB-glc). We therefore tried several alternative solutions and settled on the current representation, using a new Matlab function (<http://www.mathworks.com/matlabcentral/fileexchange/42905-break-x-axis>). If the Editors of the journal prefer the dashed-line representation, we will change the plots.

4. I am not sure that 'kinetic experiments' is the term that best describes these experiments. Consider changing to 'growth rate measurements' or the like.

Action taken: We have changed the term to "growth experiment" throughout the paper (p 5, par 2; p 7, par 2; p 12, par 2; p 17, par 2).

5. The results regarding the stability of the system (figure S16) are very important, but are buried in the discussion. These should be properly presented in the results. The authors should be cautious regarding their statements. A growth experiment of 24 or 48 hours can hardly be regarded as 'long-term'. The authors should clearly state that they observe escape from the growth-arrested state and discuss future avenues for addressing this.

Action taken: Following the suggestion of the reviewer, we have added a new subsection to the *Results*, entitled *Stability of the growth switch* (p 11-12). This subsection is based on text from the *Discussion* of the previous manuscript and Figure S16, which was moved from the Supplementary Information to the main text (where it is now called Figure 6). The latter figure was extended with

a new panel summarizing the results of an additional control experiment suggested by Reviewer 2 (see below). We removed the qualification "long-term" when referring to the 24-h and 48-h growth experiments (p 9, par 4; p 11, par 4). We clearly state that, after 48 h, some of the cultures had escaped (2/33 in the microplate experiments and 2/25 in the flask experiments) (p 12, par 2). While emphasizing that the stability of the construction meets the criterion we set beforehand, we also present some ideas to further improve it, *e.g.*, putting the expression of the RNA polymerase α subunit under the control of an inducible promoter as well (p 12, par 2).

Reviewer 2

This manuscript describes the control of E. coli growth by using inducible expression of the core RNA polymerase enzymes encoded by rpoB and rpoC. The authors showed a switch-like response of cell growth as the inducer (IPTG) concentration in the medium varied. They then showed that the non-growing cells could be revived by cycling the IPTG conc and that this switch could be operated repeatedly. Finally the system was put into a biotech application by showing that the strain with growth switch produced converted glucose to glycerol more effectively than WT cells.

Overall this is a solid piece of experimental work. Even though it is not a surprise that repressing RNAP synthesis would lead to growth arrest, doing this well is not an easy task due to the ease of mutation. The authors' use of multiple lacI genes to combat this problem is clever and effective, at least for cultures grown in 96-well microplate. However, I have some questions and concerns regarding the solidity of the specific conclusions drawn from this study.

1. The authors claim a switch-like growth transition upon RNAP titration. While the switch-like response to IPTG titration itself is clear, the authors claim that this results from a switch-like response of growth to RNAP abundance (top of p.8). This claim is based on the plot of growth rates vs RpoC-mCherry level (Fig. 3B, 3D): the data is fit by Hill functions with Hill coeff of 10. I am not at all convinced of this claim: the important region is for mCherry "concentration" from 0 to 1000 (by the scale shown in Fig. 3B, 3D). This is also the range of the RNAP concentration present in WT cells (shown by the red points.) Over this range, the experiments have only 2-3 data points, one at high growth, one near zero growth, and one in between. It is not possible to make a conclusion on the abruptness of the switch based on these 2-3 points. [Visually, one is easily biased by the data with conc \geq 1000 units; but those are irrelevant as the authors themselves stated.] Adding to this problem, I do not believe the authors growth protocol can be expected to measure slow growth accurately - to see balanced growth with a doubling time of 3-4 hours, a much longer period (say 4-5x longer than the doubling time) is needed.

In short, the author's strain and growth protocol is not designed to reveal the quantitative relation between RNAP concentration and growth. A careful characterization of this relation will likely require a very different titratable expression system and focused on the endogenous range of RNAP concentrations. This is clearly not the focus area of this study devoted to synthetic applications. But the authors should not make a claim that is outside of the means of their system.

Answer: The conclusion that the growth rate responds in a switch-like manner to the RNA polymerase concentration is not generally based on "only 2-3 data points" in Figure 3B-C. The point for 1000 μ M may not be very informative (though a useful control), but that still leaves 8

data points between 0 and 1500 fluorescence units in panels *B* (M9-glc) and *C* (M9-glc-CAA)! The reviewer is right that in panel *D* there are only 3 data points in this range, but even here the passage from 300 to 600 units leads to an abrupt rise in growth from 0 to its maximum rate. While we therefore think that taken together these data are best interpreted as showing a switch-like response of the growth rate to the concentration of RNA polymerase, we see the point of the reviewer that additional experiments may be necessary to provide a more precise quantitative characterization of the relation. Perhaps too much emphasis was given to the fact that a Hill function with coefficient 10 fits the data well. In the revised version of the manuscript, we have therefore omitted most references to this function and we have indicated that further work should unambiguously quantify the ultrasensitive dependency. The presumed ultrasensitivity is an interesting observation and a starting point for further work, but indeed not of central importance to the message of this manuscript.

Notice that all growth-rate measurements are typically based on at least 2 doubling times for growing bacteria (see the main text) and at least 200 minutes for growth-arrested cells. This latter criterion for growth-arrested cells was not clearly stated in the text and misrepresented in Figure 2*A* (where the green bar around 1100 min was much too short). With the high sampling frequency of the microplate reader (one reading every 1-2 min), this means in practice that several dozens of absorbance measurements, and much more for slowly-growing cells, were used for each growth-rate estimation. The statistics (confidence interval) of all fits of exponential growth curves (two parameters) to the absorbance data were verified and no anomalies were detected. The reported data in Figure 2 are the mean of the growth rates thus estimated for at least five independent biological replicates started from single colonies on a Petri dish (the mention of "technical replicates" in the caption of Figure 2 is incorrect and should be "biological replicates", in accordance with the correct description of the procedure in the *Materials and methods*). The small error bars in the plots, representing \pm two standard errors of the mean, make us confident that we were able to obtain highly-precise growth-rate measurements in this manner!

Action taken: We have changed the title of the subsection *Quantitative dependence of the growth rate on the concentration of RNA polymerase* to *Dependence of the growth rate on the concentration of RNA polymerase* (p 7). Moreover, we indicate that, although taken together the data strongly suggest a switch-like response of the growth rate to the concentration of RNA polymerase, more work is necessary to unambiguously **quantify** the ultrasensitive dependency (p 8, par 1; p 13, par 1). Throughout the paper, we have weakened our conclusions on this point (p 2, par 1; p 4, par 1; p 8, par 2; p 12, par 5) and the subsection *Ultrasensitive response of growth rate to changes in RNA polymerase concentration* in the *Discussion* has been removed (some of the other material in this subsection, not directly related to the point at issue, has been moved up to the *Discussion* subsection *Medium-independent and reversible growth switch*) (p 13, par 1-2). We have also added more detail on the growth-rate measurements to convince the reader of the quantitative precision of the reported values, along the lines sketched above, and we have corrected the mistakes in panel *A* and in the caption of Figure 2 (p 6, par 2; p 20, par 2).

2. *The authors explained the importance of controlling cell growth as a strategy to direct nutrient flux into the production of desired products (rather than cell mass) in biotech applications, and stated problems with previous attempts at this strategy. Based on the higher conversion efficiency of glucose to glycerol in the R-strain compared to the W-strain, the authors concluded that their inducible RNAP strain presents an effective strategy. I am not convinced of this claim for several reasons given below.*

Fig. 5A, 5B are the key plots on which the conclusions on glucose-to-glycerol conversion efficiencies are based. I see multiple deficiencies with the comparison. First the grey-shaded regions used for evaluation are not chosen on equal footing: The shaded region for strain W corresponds to exponential growth while that for strain R corresponds to transition to non-growing phase. To make comparison on equal footing, the minimum the authors should do is to arrange their culture so that the two strains have similar saturating OD, and select regions from the same OD range, e.g., from 50% saturation to 90% saturation. [In order to monitor glycerol synthesis directly for the entire range, it will be useful to remove glycerol uptake from strain W.] The comparison can be made on a more equal footing if the saturation of OD can be arranged to result from something other than glucose running out. They can, e.g., supply antibiotics to the W-strain while supplying IPTG to the R-strain, or they can apply their titratable system to control an essential metabolic gene. It is only then statement on the advantage of various growth control strategies be made.

Indeed, controlling metabolism or switching metabolic genes have been used in the metabolic engineering community in the past as the authors described in the text. However, just because the R-strain can do well converting glucose to glycerol does not mean the other strategies could not do an equally good job at this particular task. As the authors mentioned, a big problem with the strategy of controlling growth via metabolism is the cell's ability to adjust to metabolic perturbations. A specific problem is the inhibition of carbon uptake when growth is arrested by, e.g., depriving cells of supply of auxiliary nutrients such as nitrogen or sulfur. An example is the inhibition of PTS sugar uptake systems by a build-up of pyruvate. In the glucose-to-glycerol study done in this work, incoming carbon is diverted to synthesizing glycerol from DHAP, thus possibly relieving the build up of pyruvate. It is thus conceivable that other modes of metabolic perturbations can do the job equally well as controlling RNAP synthesis. The authors made no comments on why they think controlling RNAP synthesis would alleviate the cells from experiencing the same problems found by controlling metabolism.

Answer: It is important to emphasize that the experiment reported in Figure 5 did **not** aim to show that for this particular application (glycerol production from glucose) our strategy performs **better** than alternative strategies of growth arrest proposed in the literature. If this had been our claim, we should indeed have performed an experiment similar to the one proposed by the reviewer, reducing growth of the W-gly strain by other means (*e.g.*, antibiotics). Rather, the aim of the experiment was to provide a proof-of-principle that arresting growth by our strategy can improve metabolic production yields in a concrete and biotechnologically relevant application. This is not evident *a priori*, because other regulatory mechanisms in the cell could resist the desired reorientation of nutrient fluxes. With this aim in mind, the experiment in Figure 5 is the right experiment to perform, because we compare two strains that only differ in the rate of synthesis of RNA polymerase. Both strains initially have the same growth rate and glycerol production yield, but whereas one strain (W-gly) continues to grow and produce glycerol at a constant yield until glucose runs out, the other strain (R-gly, 0 μ M IPTG) slows down its growth and correspondingly increases the glycerol production yield until all glucose has been exhausted (Figure 5C-D). This result provides the desired proof-of-principle of the applicability of our approach, as suggested by the reviewers of the original manuscript. The fact that the two strains produce glycerol in different growth stages, exponential growth for the W-gly strain and (transition to) growth arrest for the R-gly strain without IPTG is essential for the proof-of-principle.

Beyond the proof-of-principle, we do see some favorable properties of the strategy of arresting growth through the control of RNA polymerase expression: the growth switch is reversible (Figure 4 and Figure EV4), not dependent on the medium composition (Figure 2 and Figure EV1),

genetically stable for at least 24 h (Figure 5), and easily connectable to other regulatory circuits in the cell (subsection *Growth switch is modular* in *Discussion*). However, the reviewer is of course right that there may be situations in which our strategy will also experience some of the problems that confront alternative approaches, *e.g.*, the perturbation of cofactor balances or the build-up of toxic intermediate metabolites. A comprehensive validation study applying the growth switch to different products and comparing the outcome with other approaches for growth arrest would be necessary to map the strengths and limitations of our strategy, but this is outside the scope of the current manuscript. Although we do not naively believe that the approach presented is a panacea, we do think it is an extremely promising extension of the toolbox of bioengineers.

Action taken: We have more clearly explained the aim of the experiment in Figure 5, and we have been more explicit in what is claimed and not claimed in both the subsection *Growth switch enables the improvement of metabolic production yields* of the *Results* (p 11, par 2) and in the *Discussion* (p 14, par 4). In particular, we explain the role of the experiment as a proof-of-principle of the interest of the growth switch for biotechnological applications, but we state that Figure 5 does not demonstrate that, for the specific problem of glycerol production from glucose, our strategy performs better than alternative, metabolic approaches of growth arrest. Moreover, we add a brief discussion on potential limitations of our approach that might possibly occur in other applications (p 11, par 2).

Finally, direct inspection of Fig. 5B shows that the 2/3 of glycerol was actually produced during the slow down of cell growth, as is known for growth control by metabolic means. Glycerol synthesis essentially stopped 300 minutes beyond the grey zone. This is at variance with the authors goal (stated in introduction) that enzymes will continue to function after switching off growth, enabling high-yield production of metabolites of interest. Perhaps this 300-minute after growth ceases is longer than what is achievable by other strategies. If so, then this should be emphasized, backed up by direct comparison in the post-growth regime.

Answer: The reviewer is right that only about 1/3 of glycerol was produced during complete growth arrest, but notice that at this stage most of the glucose had already been utilized, preventing prolonged glycerol production. However, even if the highest production yield would be obtained during the slow-down phase of growth rather than during complete growth arrest, we do not see this as a limitation of our approach, given that cells can be cycled between sequences of growth, growth slow-down, and growth arrest (Figure 4 and Figure EV4). It raises the interesting question of the dynamic optimization of growth conditions to maximize metabolic yields (Venayak *et al.*, 2015).

Action taken: We repeat the observation of the reviewer in the subsection *Growth switch enables the improvement of metabolic production yields* (p 10, par 4) and discuss its possible implications for the dynamic optimization of process conditions in the *Discussion* (p 15, par 2).

3. It is very nice to see the demonstration that cell growth can be revived after up to 8-hr of no growth when medium contained no IPTG. Microfluidic devices are appropriate for switching out of IPTG. The observation of cell elongation without division is also very interesting. For biotech applications however, it will be useful to know the stability of these cells, especially during the stressful period with no IPTG, in shake flasks where there are 1000x more cells than in 96-well microplates (and 10⁶ more than then 950 tracks followed in microfluidic devices), and hence with much higher rate of mutation to overcome the repression of RNAP synthesis.

Answer: We have done an additional control experiment in flasks to test the stability of the genetic construction and found that none of the 25 cultures had escaped after 24 h and only 2 after 48 h. These results are in excellent agreement with the results from the microplate experiment.

Action taken: We report the flask experiment in Figure 6B and in the new subsection *Stability of the growth switch* in the *Results* (p 12, par 1). The details of the experiment are described in the *Materials and methods* (p 16, par 2).

Reviewer 3

The authors have addressed any concerns I had with the previous submission. I have only very minor comments.

Minor points:

-”These experiments have shown that cell composition is correlated with the growth rate and not the nature of the specific nutrients available for growth” This is only approximately true; see Ehrenberg, Bremer and Dennis (2013) ”Medium-dependent control of the bacterial growth rate” Biochimie 95:643-58. PMID:23228516

Action taken: The reviewer is right, we have amended the phrase to indicate that the culture history may also play a role, especially when the nutritional quality is low, citing the 2013 review of Ehrenberg *et al.* (p 3, par 1).

-p3. regain – > regained

Action taken: Corrected (p 3, par 1).

-p6. ”this phenotype is therefore independent of the specific induction system used.” The control is a lacY deleted mutant still employing the lac-based expression system. The claim that the phenotype is independent of the specific induction system is strong; perhaps what was meant was ”this phenotype is therefore not attributable to positive feedback in the inducer (IPTG) uptake.”

Action taken: Corrected (p 6, par 4).

-p7. ”the sensitivity of the growth rate to the IPTG concentration is much lower at higher growth rates.” Not clear what is meant by sensitivity here. Is this the slope of the interpolating curve on a log-log plot? Fit with a Hill function, the sensitivity (i.e. Hill coefficient) appears, by eye, to be roughly the same for all curves.

Action taken: We have deleted this phrase (p 7, par 1).

-Fig 4. A scale bar in the image panels would be helpful.

Action taken: The scale bar has been added.

-p9."growing to a length of about 20 m 6 hours after IPTG removal." This extreme filamentation suggests induction of the SOS response, or a transcriptional toxin/antitoxin pair. Although not directly relevant, it should perhaps be remarked in the text that the growth arrest appears filamentous.

Answer: We had significantly reduced the discussion of filamentation in the resubmitted manuscript, following a suggestion of Reviewer 1 of the original manuscript. Filamentation may be due to the SOS response, as suggested by the reviewer, but could also be a consequence of the decrease of the concentration of a protein necessary for cell division as RNA polymerase is diluted out (and transcription of this factor stops).

Action taken: We have added a comment in the *Discussion* section that growth appears filamentous, listing some of the possible causes (p 13, par 5).

References

Venayak N, Anesiadis N, Cluett WR, Mahadevan R (2015) Engineering metabolism through dynamic control. *Curr Opin Biotechnol* **43**: 142–52

Thank you again for submitting your work to Molecular Systems Biology. We have now heard back from the referee who agreed to evaluate your manuscript. As you will see below, this referee is now satisfied with the modifications made and thinks that the study is suitable for publication.

REFEREE REPORT

Reviewer #2:

I am satisfied with the changes made.

Just have one comment for the authors to consider: having many points over a period of two doublings does not necessarily capture accurately the "true" rate of steady state growth, as a culture may take quite a few doublings to reach the steady state. In the old days, a culture was adapted for ~10 doublings in the final medium before the doubling rate was measured.